# A segmentation approach for the reproducible extraction and quantification of knickpoints from river long profiles

Boris Gailleton[1], Simon M. Mudd[1], Fiona J. Clubb[2], Daniel Peifer[3], and Martin D. Hurst[3]

[1]School of GeoSciences, University of Edinburgh, Drummond Street, Edinburgh EH8 9XP, UK
[2]Institute of Earth and Environmental Science, University of Potsdam, 14476 Potsdam-Golm, Germany
[3]School of Geographical and Earth Sciences, University of Glasgow, University Avenue, Glasgow G12 8QQ, UK

*Correspondence to:* Boris Gailleton (b.gailleton@sms.ed.ac.uk)

**Abstract.** Changes in the steepness of river profiles or abrupt vertical steps (i.e. waterfalls) are thought to be indicative of changes in erosion rates, lithology, or other factors that affect landscape evolution. These changes are referred to as knickpoints or knickzones and are pervasive in bedrock river systems. Such features are thought to reveal information about landscape evolution and patterns of erosion, and therefore their locations are often reported in the geomorphic literature. It is imperative that studies reporting knickpoints and knickzones use a reproducible method of quantifying their locations, as their number and spatial distribution play an important role in interpreting tectonically active landscapes. In this contribution we introduce a reproducible knickpoint and knickzone extraction algorithm that uses river profiles transformed by integrating drainage area along channel length (the so-called integral or $\chi$ method). The profile is then statistically segmented and the differing slopes and step changes in elevations of these segments are used to identify knickpoints and knickzones, and their relative magnitudes. The output locations of identified knickpoints and knickzones compare favourably with human mapping: we test the method on Santa Cruz Island, CA, using previously reported knickzones and also test the method against a new dataset from the Quadrilátero Ferrífero in Brazil. The algorithm allows extraction of varying knickpoint morphologies, including stepped, positive slope-breaks (concave upward) and negative slope-break knickpoints. We identify parameters that most affect the resulting knickpoint and knickzone locations, and provide guidance for both usage and outputs of the method to produce reproducible knickpoint datasets.

## 1 Introduction

Landscapes are shaped by competition between crustal processes such as tectonic plate motion or dynamic topography and deposition or erosion at the Earth's surface. This competition, if unperturbed, tends toward topographic steady-state where vertical motions are counterbalanced by erosion (e.g., Hack, 1960; Willett and Brandon, 2002). In unglaciated landscapes, the main driver of erosion is the river system (Ahnert, 1970), which incises the landscape to remove and transport material from uplands to active basins. The analysis of river long profiles has been a key method to interpret landscape evolution (e.g., Wobus et al., 2006), from the early recognition of graded rivers (e.g., Gilbert, 1877) to the generalised recognition that river profiles reflect varying erosion processes (e.g., Mackin, 1948; Hack, 1960; Howard, 1965; Howard et al., 1994; Dietrich et al., 2003; Kirby and Whipple, 2012).

In a river system, topographic steady state requires spatially stable rock uplift and climatic conditions over a long period of time (Willett and Brandon, 2002). In most landscapes, however, these conditions are unlikely (Baldwin et al., 2003). Many processes have been suggested to result in both spatial and temporal variations in uplift rate, such as varying tectonic stress (e.g., Kirby and Whipple, 2012), complex mantle processes inducing vertical motions (e.g., Faccenna and Becker, 2010; Braun, 2010), uplift driven by differential rock density (Braun et al., 2014) and base level variations linked to eustatic variations (e.g., Powell, 1875; Lambeck and Chappell, 2001; Schumann et al., 2016). River systems affected by these processes respond by transmitting signals upstream through the channel network (e.g., Whipple et al., 1999; Royden and Perron, 2013), eventually driving drainage network reorganisation resulting in additional transient signals (e.g., Mather, 2000; Castelltort et al., 2012; Willett et al., 2014; Whipple et al., 2017b; Mudd, 2017). Moreover, river profiles are also affected by intrinsic landscape properties, such as fracture density (e.g Whipple, 2002) or differential lithology (e.g., Stock and Montgomery, 1999; Forte et al., 2016) which can also lead to morphological adjustment of the channel (e.g., Kirby and Whipple, 2012). The most direct and widely observed expression of river adjustment to transient or intrinsic perturbations is a discrete change in river gradient, commonly referred as a "knickpoint".

Changes in channel gradient linked to different lithologies have been recognised in geomorphological studies for centuries. Lapparent (1896) suggested that these changes may represent "successive reaches" with different base levels, hypothesising that these reaches somehow migrate upstream. Davis (1889) recognised the tectonic genesis of some of these signals, describing how landscapes experience erosion cycles with periods of "rejuvenation" followed by periods of gradual adjustment, and thus transience. However, these early studies did not name such morphologies as distinct entities. The term "knickpoint" was first introduced into the geomorphological literature by Knopf (1924), borrowing the word from chemical sciences to "denote an abrupt change in direction from a gentle concave curve to a curve that is convex upward" (p.636).

Based on earlier observations on the topography and geology of the Appalachians (e.g., Barrell, 1920; Bascom, 1921), Knopf (1924) described a knickpoint as a migrating steepened boundary between two river reaches. She went on to state that the downstream reach should flow with a gradient determined by the present day balance between uplift and erosion, and the upstream reach should flow with a gradient representing an older such balance. Recognition of knickpoints and their significance in transient landscapes has driven much research into interpreting topography (e.g., Wobus et al., 2006; Crosby and Whipple, 2006; Abbühl et al., 2011; Kirby and Whipple, 2012), as well as using river profiles to extract past uplift histories (e.g., Pritchard et al., 2009).

The diverse nature of knickpoint formation means that these features have been used to investigate many geomorphological problems. For example, retreat rates have been used to link knickpoints with tectonic events and faulting (e.g Attal et al., 2008, 2011; Whittaker and Boulton, 2012) or climatically triggered base-level fall (e.g., Crosby and Whipple, 2006; Baynes et al., 2015; Neely et al., 2017). Although migrating knickpoints are commonly associated with base level variations, Haviv et al. (2010) highlighted the role of differential lithologies in retreat rates of vertical knickpoints within tectonically and climatically stable landscapes. Furthermore, Scheingross and Lamb (2016) and Scheingross et al. (2017) noted the importance of sediment supply and hydraulic conditions in waterfall retreat, providing a quantitative interpretation of the early observations of Lapparent (1896) on waterfall migration. Cook et al. (2013) observed an important correlation between knickpoint retreat

and bedload transport, further highlighting the importance of sediment transport. Bishop and Goldrick (2010) demonstrated that considering the role of resistant lithologies is crucial when studying landscape evolution, as they can considerably slow down landscape response time to transient signals. Other studies have linked knickpoints directly to landscape characteristics such as heterogeneous lithology (e.g., Tucker and Slingerland, 1996; Stock and Montgomery, 1999; Kirby et al., 2003; Duvall, 2004). Recent analogue experiments on knickpoint retreat (e.g., Baynes et al., 2018) have highlighted the inter-connectivity of all these processes and the need to consider both internal and external landscape characteristics.

These examples demonstrate the importance but also the diversity of transient and lithologic signals in landscapes, and highlight that different processes can generate remarkably similar channel morphology. It is therefore crucial to define knickpoints morphologically before drawing interpretations about their significance in term of processes or genesis. In this contribution, we aim to provide a method for reproducibly and systematically extracting knickpoints within real landscapes based on river profile morphology.

## 1.1 Knickpoint morphology and detection

### 1.1.1 Morphological description

Knickpoints can be defined as discrete changes in river gradient (Whipple et al., 1999). Haviv et al. (2010) proposed two end-member knickpoints: break-in-slope knickpoints, expressed by an abrupt change in river gradient; and break-in-elevation knickpoints, characterized by step in the elevation as a waterfall with similar gradient on both sides of the knickpoint. These knickpoints are now commonly referred as slope-break knickpoints and vertical-step knickpoints (e.g., Kirby and Whipple, 2012; Neely et al., 2017). Kirby and Whipple (2012) suggest that although vertical-step knickpoints tend to be linked to discrete heterogeneities along the river profile (e.g., caused by geological boundaries), both morphologies can be either fixed or mobile and each style of knickpoint may be generated by a range of processes.

As discussed in Goldrick and Bishop (2007) and Kirby and Whipple (2012), both morphologies can be detected using a slope–area plot (Figure 1) or a slope–distance plot. It has long been observed that channel gradients vary systematically as a function of drainage area. For example, Gilbert (1877) stated that "In general we may say that, *ceteris paribus*, declivity bears an inverse relation to quantity of water (p.114)." How do we then find anomalous channel gradients? In the mid-twentieth century, authors such as Hack (1957) and Morisawa (1962) found systematic, quantitative relationships between channel gradient and drainage area, often used as a proxy for discharge. Morisawa (1962) and later Flint (1974) recognised that channel gradients often declined systematically downstream in a trend that could be described by a power law:

$$S = k_s A^{-\theta}, \tag{1}$$

where $\theta$ is referred to as the concavity index since it describes how concave a profile is: the higher the value, the more rapidly a channel's gradient decreases downstream. The term $k_s$ is called the steepness index, as it sets the overall gradient of the channel, and a number of authors have noted that $k_s$ frequently scales with erosion rate in lithologically homogeneous

landscapes (e.g., Ouimet et al., 2009; DiBiase et al., 2010; Scherler et al., 2014; Mandal et al., 2015; Harel et al., 2016). A knickpoint might manifest itself as an abrupt change in slope–area scaling, and lead to local variations in $k_s$ (Figure 1a).

However, using slope–area data derived from digital elevation models (DEMs) suffers from noise in channel slopes, leading to scattering of gradient data, as discussed in Perron and Royden (2013). Wobus et al. (2006) proposed methods to reduce the effect of noise and extract trends from slope–area plots. These recommendations include regular sampling of elevations to extrapolate artefact-free contour lines or logarithmic binning by drainage area. Smoothing induces inexorable data loss and may result in difficulties detecting subtle, but important features such as knickpoints (Figure 1b).

Alternatively, we can integrate equation (1), since $S = dz/dx$ where $z$ is elevation and $x$ is distance along the channel (e.g., Whipple et al., 2017a), resulting in

$$z(x) = z(x_b) + \left(\frac{k_s}{A_0{}^\theta}\right) \int_{x_b}^{x} \left(\frac{A_0}{A(x)}\right)^\theta dx, \qquad (2)$$

where $A_0$ is a reference drainage area, introduced to nondimensionalise the area term within the integral in equation (2). We can then define a longitudinal coordinate, $\chi$ (Royden et al., 2000):

$$\chi = \int_{x_b}^{x} \left(\frac{A_0}{A(x)}\right)^\theta dx. \qquad (3)$$

$\chi$ has dimensions of length, and is defined such that at any point in the channel

$$z(x) = z(x_b) + \left(\frac{k_s}{A_0{}^\theta}\right)\chi. \qquad (4)$$

The $\chi$ approach to represent normalised long profiles (equations (4) and (3)) can serve as an alternative method to explore the slope–area relationship within a drainage network. The $\chi$ coordinate integrates information about drainage area, while requiring less smoothing and lumping than $\log(S)$–$\log(A)$ plots (Figure 1c). This approach has been widely used in recent studies (e.g., Perron and Royden, 2013; Mudd et al., 2014; Willett et al., 2014; Mouchené et al., 2017; Whipple et al., 2017b; Neely et al., 2017; Moodie et al., 2017).

### 1.1.2 Existing algorithms

Traditional knickpoint identification from DEMs relied upon user-based selection along river long profiles (e.g., Hayakawa and Oguchi, 2006; Wobus et al., 2006). Several computational methods have been proposed for extracting knickpoints from DEM-derived datasets. The first (semi-)automated methods taking advantage of digital topographic data used long-profile geometry to isolate knickpoints or knickzones. Hayakawa and Oguchi (2006) proposed a semi-automated extraction method based on decreasing of gradient with increasing length. This method involved the use of ArcGIS and spreadsheet software to process

the outputs for each river. Recognising the need for automated regional knickpoint mapping methods in geomorphological studies, Gonga-Saholiariliva et al. (2011) proposed an automated algorithm to map abrupt changes in river gradient using slope, profile, and planview curvature. Gallen et al. (2013) used systematic changes in profile convexity over given thresholds ($> 20$ metres in elevation drop, coupled with a slope threshold $\geq 0.1$) to isolate knickpoints in fluvially-dominated channels with the aim of reconstructing rejuvenation events, both climatically and tectonically driven, in the southern Appalachians. A similar method has been implemented in ArcGIS by Queiroz et al. (2015). More recently, Zahra et al. (2017) published an ArcGIS toolset (called KET) that automates and optimizes the Hayakawa and Oguchi (2006) method. These methods are based of the direct use of channel elevation, gradient and curvature, and so are susceptible to previously described limitations related to noise. Furthermore, the Hayakawa and Oguchi (2006) method does not incorporate drainage area information, which is an important parameter to consider when studying knickpoints over large spatial scales, or when interpreting the retreat rates of these features.

Another set of methods exploit the use of $k_s$ from equation 1 (or $k_{sn}$ when calculated using a fixed value of $\theta$) to extract knickpoints from slope-area plots, as reviewed by Neely et al. (2017). These methods suffer from limitations linked to slope-area scattering, noise sensitivity and difficulty in precisely locating knickpoints because of the stepped nature of drainage area (increasing instantaneously downstream when a new tributary reaches the river channel). To ameliorate problems with noise and data scattering, Bennett et al. (2016) devised a method that first calculates $k_{sn}$ on channel profiles smoothed using the algorithm of Schwanghart and Scherler (2014). This derives $k_{sn}$ either from regression of slope–area plots, or using the first-order derivative of $\chi$ plots. The method selects a knickpoint where the ratio between downstream and upstream $k_{sn}$, averaged with two 2-km long serial windows, exceeds a factor of two.

Neely et al. (2017) developed an algorithm focused on knickzone detection (KZ-picker). Knickzones are selected from normalized profiles (using the approach of Perron and Royden (2013)) by comparison with a reference profile, calculated for a defined concavity index ($\theta$ in equation 1). This reference profile is a line in $\chi$–elevation space between the outlet and headwaters of the channel, and knickzones are then defined based on the deviation of the $\chi$ profile from the reference. After initial detection, knickzones are quantified by their relief (elevation drop) and adjusted using several filters or lumping-window parameters. This method is well adapted to detect knickzones that are composed of a base and a lip separating a steepened reach. Example of output produced by this algorithm and compared to our is presented in section (5.4).

Another method for extracting knickpoints has recently been implemented using TopoToolbox (Schwanghart and Scherler, 2014). Albeit unpublished, the code is available and also aims to reproducibly extract knickpoint locations from river profiles. It selects knickpoints by creating reference channel profiles that are concave up and then selecting knickpoints where the actual channels are the most different from the reference channels. Although not based on the slope–area relationship, this method is perhaps the closest algorithmic attempt to match the knickpoint definition of early workers (e.g., Knopf, 1924). A sensitivity parameter defines the number of iterations and indirectly the number of knickpoints detected. After knickpoint extraction, a value is attributed to each identified knickpoint quantifying the divergence of the long profile from the reference profile. We discuss the similarities and differences of this method compared to our method in Section 6.

### 1.1.3 Motivation for a new method

Despite the large number of past approaches to selecting knickpoints, we have developed a new method because i) many authors still select knickpoints based on qualitative interpretation of channel long profiles or slope–area data and we desired an open source, reproducible method that has no reliance on proprietary software such as ArcGIS (e.g., Hayakawa and Oguchi, 2006) or MATLAB (e.g., Schwanghart and Scherler, 2014; Neely et al., 2017); ii) channel erosion is modelled to scale with discharge, and therefore we wished to use a method that includes discharge (or its proxy drainage area); iii) existing slope–area approaches make it difficult to pinpoint knickpoint locations (Figure 1), and therefore we choose to use a $\chi$-based approach; iv) we wished to develop a method that not only selected knickpoint locations but included metrics of changes in normalised channel steepness, as that metric is frequently used in tectonic geomorphology and v) we aimed to create a method allowing the differentiation between different knickpoints morphologies (e.g., slope-break vs vertical-step).

Although the newest methods (Schwanghart and Scherler, 2014; Neely et al., 2017) meet a subset of these criteria, they both only describe a specific morphology of knickpoint/knickzone and use indirect methods to quantify their magnitude (e.g. derived from the comparison with a reference profile). Our aim here is to provide a method that selects locations, styles (e.g. vertical step, slope-break), and magnitudes (e.g. main features or secondary ones) of knickpoints and knickzones that is free of manual selection in order to complement these existing methods that are more focused on identifying locations of a particular style of knickpoints and knickzones (e.g. waterfall).

We provide comparisons with two existing methods in section 5.4. These have been chosen for the following reasons: i) the knickpoint-extracting algorithms are open-source (with the limitation of MATLAB licenses), ii) the methods are objective, reproducible and provides a quantification of knickpoint magnitude in order to compare it with our and iii) (Schwanghart and Scherler, 2014) is purely based on channel morphology while (Neely et al., 2017) uses the slope–area relationship and $\chi$ thus providing a reasonable comparison of our algorithm with the range of existing methods.

## 2 Methods

An overview of our knickpoint identification method can be found in figure 2.

### 2.1 DEM preprocessing and river network extraction

Firstly, we fill the DEM using the filling algorithm of Wang and Liu (2006), to make sure that each cell has a flow direction and to avoid internal basins generated by DEM noise (e.g., Barnes et al., 2014). This approach is suitable for cases where no feature is spuriously damming the DEM. Spurious damming can occur when vegetation, bridges, or other features lead to high elevations over the channel when in fact the channel sits at a lower elevation. The filling process will create flat surfaces behind such spurious dams and will therefore hinder channel profile analysis.

If features that lead to spurious damming are present, we give users the option to use a breaching or carving algorithm. This excavates through spurious dams to avoid overfilling. The depression-breaching algorithm in our code is that created

by Lindsay (e.g., 2016) and adapted from Barnes (2016) within our method. It is also possible to supply the algorithm with preprocessed DEMs (e.g., Schwanghart and Scherler, 2017).

From the preprocessed, carved, or filled DEM, we provide several methods of extracting the river network, including the DrEICH method (Clubb et al., 2014); a curvature method proposed by Pelletier (2013); and a method that uses a Wiener filter (Wiener, 1949) that combines elements of the methods of Pelletier (2013) and Passalacqua et al. (2010) first implemented by Grieve et al. (2016) and Clubb et al. (2017); Grieve et al. (2016) found this latter method least sensitive to DEM resolution. Finally, we include extraction based on a drainage area threshold, more suitable for low-resolution DEMs (e.g., SRTM, ASTER) or large-scale studies where the location of channel heads is less important. We also ensure during the preprocessing that no catchments are beheaded by the edge of the DEM, as the $\chi$ coordinate is a function of drainage area and therefore incomplete basins will have incorrect $\chi$ values.

## 2.2 $k_{sn}$ extraction

Following channel extraction, we then calculate the $\chi$ coordinate for the resulting network. A key parameter that must be constrained prior to calculation of $\chi$ is the concavity index ($\theta$). Changing the concavity index significantly affects values of the the $\chi$ coordinate (e.g., Kirby and Whipple, 2012; Gasparini and Whipple, 2014; Mudd et al., 2018) and therefore subsequent knickpoint extraction. We select the concavity index using a method developed by Mudd et al. (2018). This method calculates the $\chi$ coordinates for a range of concavities within each watershed, and determines the most likely concavity index by directly comparing the collinearity of points on each tributary with the trunk channel (Perron and Royden, 2013; Mudd et al., 2018). This approach does not assume linearity in $\chi$–elevation space, and therefore is applicable in transient landscapes (Mudd et al., 2018).

Once we determine $\theta$ values for each basin, we calculate $\chi$ and then use $\chi$–elevation profiles to determine changes in $k_{sn}$, which is the gradient of the $\chi$–elevation profile when we set $A_0 = 1$ (see equation 2). Theoretical work by Royden and Perron (2013) suggested that in eroding landscapes changes in erosion rates would be represented by changes in $\chi$–elevation gradient between segments of channels that would be linear in $\chi$–elevation space, which Royden and Perron (2013) called slope patches. Mudd et al. (2014) devised a statistical method that identified the most likely linear segments in $\chi$–elevation space. This technique searched all possible combinations of channel pixels and used the corrected Akaike Information Criterion (AICc) (Akaike, 1974; Hurvich and Tsai, 1989) to balance goodness of fit of linear segments against over-fitting the data. Here we use this same algorithm to search for breaks in slope within the profile corresponding to knickpoint locations.

Knickpoints will manifest themselves as changes in the slope of these patches, equivalent to the slope-break knickpoints of Kirby and Whipple (2012), whereas knickzones will be represented by patches with locally high gradients. That is, knickpoints and knickzones result in either changes in or locally high values of $k_s$ (or $k_{sn}$ if calculated with a fixed concavity index). The segmentation algorithm casts the profile as a series of linear segments, and each segment has a gradient and an intercept. The gradient reflects $k_s$ of the segment and the intercept can be used to detect vertical-step knickpoints, as it detects elevation jumps between adjacent slope patches.

The method developed by Mudd et al. (2014) subsamples underlying topographic data iteratively: on each iteration nodes from the channel network are chosen randomly and segmentation is applied to this subset of nodes. The number of iterations is called $n_{MC}$. This iterative approach was taken because it significantly reduces the sensitivity of the results to user parameters (Mudd et al., 2014). The computational expense of the segmentation scales highly nonlinearly with the number of nodes so channel profiles are broken into subsections of length $n_{tg}$ (called the "Target Nodes" in Mudd et al. (2014)). The sampling of the underlying data on each iteration is random: after each sample nodes are "skipped" randomly, the number of nodes skipped varies with a uniform distribution from zero to twice a parameter $n_{sk}$ such that the mean "skip" is $n_{sk}$. We explore the sensitivity of the method to these parameters in the discussion.

The final $k_{sn}$ values are an average of many iterations using different channel profiles subsampled from the raw data, as are intercepts of local segments. These averaged values are used to build segmented elevation. Each node then represents an average of the best-fit segments for every iteration of the segmentation routine (Figure 3a):

$$z_{segi} = M_{\chi i} * \chi_i + b_{\chi i}, \tag{5}$$

where $i$ is the given node, $z_{seg}$ its elevation on the segment, $M_\chi$ the average gradient of the segments and $b_\chi$ the averaged intercept of the segments. $M_\chi$ can be expressed with the following equation:

$$M_\chi = (\frac{E}{K * A_0^m})^{1/n}, \tag{6}$$

We note here that $M_\chi$ is the same as $k_{sn}$ if $\chi$ is calculated using $A_0 = 1\text{m}^2$.

## 2.3 Knickpoint extraction from $k_{sn}$ data

### 2.3.1 Change point detection

Change point detection is a common technique used within many fields (e.g., time series analysis) and a number of statistical tools have been developed to identify change points, reviewed and described by Truong et al. (2018). In our case, the signal ($k_{sn}$) is by definition piecewise stationary, and abrupt changes occur between each segment (*i.e.*, knickpoints). Change point detection algorithms aim to estimate and isolate the exact location of these boundaries between stationary patches. Method choice depends on the nature of the original dataset (*e.g.,* noise intensity) and the number of changes we aim to extract (*e.g.,* predetermined or unknown). In our case, although the segmentation algorithm of Mudd et al. (2014) can result in very sharp segment boundaries, in many cases the transitions between segments is fuzzy. We therefore have an unknown number of change points to detect from a variably noisy signal. We therefore choose to use a signal processing filter (Condat, 2013) to flatten the piecewise $k_{sn}$ patches and discretise all potential change points. This algorithm identifies where $k_{sn}$ and elevation are statistically varying the most within any transition zones. It also combines segments that have very small changes in $k_{sn}$ relative to the noise in the data (Figure 3b).

We denoise the data using a one dimensional Total Variation Denoising (TVD), a signal processing filter adapted from a optimized algorithm by Condat (2013) solving the following equation:

$$\underset{x \in \Re^N}{minimize} \frac{1}{2} \sum_{k=1}^{N} |y[k] - x[k]|^2 + \lambda \sum_{k=1}^{N-1} |x[k+1] - x[k]|, \tag{7}$$

where $N$ represents the number of samples (nodes) per population (in this case a river channel from source to next higher-order stream, or the outlet), $y$ represents the raw signal $y_1, y_2, y_3, ... y_N$, in this case $k_{sn}$ ordered by ascending $\chi$ within each river, $x$ the denoised signal $x_1, x_2, x_3, ... x_N$, referred as denoised $k_{sn}$, and $\lambda$ is a regularization parameter (Condat, 2013). This method minimises variations, where the parameter $\lambda$ must be real and greater than zero. Greater $\lambda$ values result in less variation in the processed signal, and $\lambda \to +\infty$ results in no variation in the processed signal whatsoever. The selection and sensitivity of this parameter is discussed in Section 5.1.

After denoising the data, our method then iterates through all nodes in each channel and identifies change points as any variation in the denoised $k_{sn}$ data. These represent first-order knickpoints that we quantify by their change in denoised $k_{sn}$, which we call $\Delta k_{sn}$. $\Delta k_{sn}$ is a quantitative measure of the magnitude of the slope-break component of the knickpoint (Figure 4a). We refer to change points as knickpoints in the rest of the manuscript.

### 2.3.2   Combining knickpoints

Denoised $k_{sn}$ data can still contain closely clustered steps in $k_{sn}$ values, which may in fact represent a single knickpoint. We therefore use an algorithm to determine which of these clusters can be combined. Iterating through each river, the algorithm tests the neighbouring nodes of each raw knickpoint in a window that we call the "combining window". If two knickpoints in the denoised $k_{sn}$ data are within the combining window and both have the same sign of $\Delta k_{sn}$, the two knickpoints are merged and their magnitude summed. This process is repeated using newly merged knickpoints until no nodes are within the combining window, or until a change in knickpoint sign (Figure 4b). The combined knickpoint is then centred between the combined nodes. The width of the combining window (which we denote $r_{comb}$, and is defined by a number of nodes rather than a flow distance) is a user-defined parameter, the selection of which we address in Section 5.1.

### 2.3.3   Vertical-step knickpoint detection

Small variations between segments with similar $k_{sn}$ values may be ignored by denoising, which may seem trivial if the aim is to isolate the main variations in channel steepness. However, this may lead to vertical-step knickpoints being missed if channel segments above and below the vertical-step knickpoint have similar $k_{sn}$ values despite a jump in $z_{seg}$. We therefore use a second approach to extract knickpoints, allowing us to identify both slope-break and vertical-step knickpoints.

The algorithm calculates changes in $z_{seg}$ using equation (5) in order to isolate the main jumps in profile elevation. We differentiate this value along the river nodes ($\Delta z_{seg}$) to detrend the elevation signal and focus on the stepped variations. For each node in the channel, the mean and standard deviation of $\Delta z_{seg}$ is calculated within a window of surrounding nodes; the

window width in nodes is called $r_W$. The nodes within the first and last half-windows are calculated using respectively the first and last window. $\Delta z_{seg}$ is then compared to the standard deviation of the nodes within the corresponding window multiplied by a coefficient (which we call $T_\sigma$), and the node is selected as a vertical-step knickpoint if $\Delta z_{seg}$ is greater (Figure 5b). This approach ensures that the selected vertical-step knickpoints show an anomalous increase in elevation. The selection of the window width and the coefficient are discussed in Section 5.1. We can then use $\Delta z_{seg}$ as a quantitative measure of the size of each vertical-step knickpoint.

## 2.4 Accuracy metrics

The accuracy of the method is assessed using a true positive (TP), false positive (FP) and false negative (FN) approach. This comparison method is often use to test algorithm performances on point data, such as channel heads (e.g., Orlandini et al., 2011; Clubb et al., 2014) or knickzone locations (e.g., Neely et al., 2017). We test the algorithm with these accuracy metrics using two sites where locations of hand-picked knickpoints based on field observations and river profiles are available. Knickpoints were identified at Santa Cruz Island (California, USA) by Neely et al. (2017), and we introduce a new dataset in the Quadrilátero Ferrífero, Minas Gerais, Brazil.

We define as TP a reference knickpoint detected by the algorithm, as FP a knickpoint detected by the algorithm that is not a reference knickpoint, and as FN reference knickpoints not detected by the algorithm. Neely et al. (2017) proposes a fourth kind of prediction called "mixed" to assess the knickzone base and lip detection, where only one of the two knickzone boundaries is detected. We chose not to use this approach as we define a knickpoint as a point location showing an increase or decrease of $k_{sn}$ or $\Delta z_{seg}$, which is more applicable to varying knickpoint morphologies. The definition of the different knickpoint predictions allows the calculation of sensitivity, $s$, reliability, $r$, and metrics. We also add an overall quality metric, $q$, described in Heipke et al. (1997). The sensitivity can be expressed as:

$$s = \frac{\sum TP}{\sum TP + \sum FN}, \tag{8}$$

where $\sum TP$ and $\sum FN$ are the sum of $TP$ and $FN$. This metric measures the method's ability to detect knickpoint that a user would have manually picked. $s = 1$ implies the detection of all the locations of reference knickpoints. The reliability can be expressed as:

$$r = \frac{\sum TP}{\sum TP + \sum FP}, \tag{9}$$

where $\sum TP$ and $\sum FP$ are the sum of $TP$ and $FP$. This metric measures the occurrences where the method identifying knickpoints that a user would not have picked. The overall quality metric can be expressed as:

$$q = \frac{\sum TP}{\sum TP + \sum FP + \sum FN}. \tag{10}$$

A $q$ value of unity implies perfect agreement between algorithmically and hand-picked knickpoints. We focus on these metrics instead of the knickpoint magnitude, as it is more difficult to predict and is dependent on many parameters within the extraction of the $\Delta k_{sn}$ values.

## 3 Test locations

In order to test the performance of our method, we extract knickpoints from two field sites with independently-mapped knickpoint and knickzone locations. The first of these sites is Smugglers Basin on Santa Cruz Island (California, US), where knickpoints and knickzones were mapped by Neely et al. (2017) using a combination of fieldwork and supervised selection from river long profiles. Smugglers basin is undergoing transient adjustment to climatic and tectonic signals (Neely et al., 2017). The second field site is located in the Quadrilátero Ferrífero (Minas Gerais, Brazil), where we present a new dataset of extracted knickpoint and knickzone locations from field observations and river profiles. Quadrilátero Ferrífero represents a more stable site in term of climate and tectonics (e.g., Dorr, 1969; Salgado et al., 2008), and therefore knickpoints in this landscape have been linked instead to changes in lithology.

### 3.1 Santa Cruz Island, USA

The first calibration test site is the headwaters of the Smugglers Cove catchment, located in the SE of Santa Cruz Island, the largest of the California Channel Islands (California, USA). Lidar data at 1 m resolution are available in the basin via the 2010 US Geological Survey Channel Islands lidar Collection, available from OpenTopography (opentopography.org).

The basin has a total relief of approximately 550 m and drains to the Pacific Ocean. Previous work has estimated uplift rates of $\approx$1 mm yr$^{-1}$ using dated terraces and fault activity (e.g., Pinter et al., 1998; Muhs et al., 2014), and the site has experienced regional sea-level variations (e.g., Schumann et al., 2016; Pinter et al., 2018). This, along with bedrock heterogeneity, has led to numerous knickzones in the catchment which have been mapped and tested against a previous knickzone extraction algorithm by Neely et al. (2017). 18 knickzone bases and lips have been reported based on topographic expression and field observations across the whole catchment. As the Neely et al. (2017) algorithm is targeted specifically at knickzones, we compare the mapped knickzone bases and lips with those picked by our algorithm. Knickzone bases and lips are the equivalent of increases and decreases in $k_{sn}$, respectively.

We extracted channel heads using a curvature-based method of channel extraction, following Pelletier (2013) and Grieve et al. (2016). This method has an estimated accuracy of $\approx$10 metres horizontally along drainage paths (Clubb et al., 2014). Before extracting channel steepness, we calculated the best fit concavity index for the basin by maximising collinearity between the main stem channel and the tributaries in $\chi$–elevation space, using the bootstrapping method of Mudd et al. (2018): the best-fit $\theta$ at the site is 0.25.

## 3.2 Quadrilátero Ferrífero, Minas Gerais, Brazil

The second calibration test site is located in the eastern part of the Quadrilátero Ferrífero (QF, Brazil), in a basin draining the Caraça Range (Figure 8). The QF is an area of relatively high elevation in southeastern Brazil, and the Caraça Range is its most pronounced topographic feature with a maximum elevation of $\approx$2100 m and maximum relief of $\approx$1500 m. Tectonic activity is thought to have ceased by $\approx$500 Ma (e.g., Dorr, 1969; Chemale et al., 1994; Alkmim and Marshak, 1998). Upstream areas are primarily underlain by resistant rocks (e.g., quartzites and banded iron formations), whereas less resistant rocks often underlie downstream areas (e.g., schists and phyllites). The association of mountainous topography and long-term tectonic stability have led to controversy in the post-orogenic evolution of the QF (Peifer Bezerra, 2018). The most accepted hypothesis is that differential denudation of lithologies with different resistance to denudation has led to a geomorphic differentiation where the uplands, underlain by strong rocks, are high because they have been denuded less and more slowly than their surroundings (e.g., Harder and Chamberlin, 1915; James, 1933; Varajão, 1991; Salgado et al., 2008; Peifer Bezerra, 2018). An alternative hypothesis is that the relief of the QF results from a complicated history of geographic cycles interrupted by epeirogenic uplift (e.g., King, 1956; Dorr, 1969; Barbosa, 1980).

Knickpoints are common features in the rivers flowing away from the Caraça Range (Figure 8). These rivers have headwaters at high elevations ($\approx$2000 m), and their long profiles display many convexities associated with substantial elevation drops (up to 1.4 km of descent over $\approx$15 km of downstream distance), and steep channel and hillslope gradients. These rivers flow over quartzite terrains, yet transitioning in their distal part to schists (see Supplementary Materials 5.2). The origin of these knickpoints is unresolved, being possibly the result of spatial variations in rock resistance, or alternatively resulting from transient uplift signals that have failed to progress beyond quartzite units (Peifer Bezerra, 2018). We used a TanDEM-X DEM with 12 m resolution to extract knickpoints from the QF. Before extracting channel steepness, we estimated the best fit concavity index as 0.15 using the methods presented in Mudd et al. (2018).

## 4 Results

### 4.1 Performance at Santa Cruz Island

We carried out knickpoint extraction on Santa Cruz Island initially with parameters detailed in Table 1; the full parameter file is available in the Supplementary Materials. As explained in Section 2, extraction prior to post-processing thinning generates a dense dataset of knickpoints both within and outside knickzones identified by the calibration dataset (see Supplementary Materials 5.1). Therefore, we apply a threshold approach to thin the dataset by removing small knickpoints. We set cut-off values of $|\Delta k_{sn}| > 0.8$ and $\Delta z_{seg} > 2.1$, where knickpoints smaller than these thresholds are ignored. These values are set for this case study with the specific aim to isolate the main knickpoints while matching with the calibration dataset. This approach is fully reproducible and does not involve manual picking of knickpoints.

Our thinning procedure reduced the number of slope-break knickpoints from 398 to 160; and the number of vertical-step knickpoints from 40 to 17. This is a relatively high number of knickpoints compared to the calibration bases and lips (18 pairs).

However, this disparity can partly be explained by the differences in methods: our algorithm details discrete changes in channel morphology whereas the calibration knickzones are identified over longer channel reaches. Therefore, one mapped knickzone may contain several algorithmically identified knickpoints.

Neely et al. (2017) propose an error radius of 50 metres around each base and lip in order to test the performance of their algorithm: we used the same approach when comparing our extracted knickpoints to the calibration data. A TP is determined as any knickpoint within the calibration knickzone or the corresponding 50 m radius. A FP is determined as any knickpoint which does not lie within this radius, and a FN is determined as a base or a lip which is not identified by our algorithm. The reliability, sensitivity, and overall quality metrics are presented in Table 2. High sensitivity ($s = 0.93$) but lower reliability ($r = 0.53$) and overall quality ($q = 0.51$) suggest that the algorithm detect the bulk of human selected knickpoints, but also a significant amount of other knickpoint features. The implications of these results are discussed below.

## 4.2    Performance at Quadrilátero Ferrífero, Minas Gerais, Brazil

The application of our method in the Ribeirão Caraça basin resulted in a dense dataset of knickpoints (n = 252); see Table 1 for parameter values and the supplementary materials for full parameter file. To thin this dataset, we removed knickpoints with attributes lower than the cut-off values of $|\Delta k_{sn}| > 0.8$ and $\Delta z_{seg} > 2.1$ for the slope-break and vertical-step knickpoints respectively. This filtering procedure decreased the number of slope-break knickpoints from 252 to 108, whereas the number of vertical-step knickpoints diminished from 44 to 23. We tested the performance of our method compared to human-selected knickpoints for the Ribeirão Caraça basin using the metrics TP, FP and FN (Table 3). We used the same error radius as was used on Santa Cruz Island for consistency. These metrics (see Section 2.4) indicate that the sensitivity of our method is high for the Ribeirão Caraça basin ($s = 0.89$), and thus the bulk of human-selected knickpoints are captured by our algorithm. On the other hand, the reliability ($r = 0.60$) and the overall quality ($q = 0.56$) are lower because the number of false positives is high, indicating that our algorithm determines a relatively high number of knickpoints compared to human selection. In summary, our algorithm captures knickpoints that are visually selected for the Ribeirão Caraça basin, as well as many knickpoints that are not recognised by traditional field mapping of knickpoints, but are morphologically similar, as defined by our algorithm.

## 4.3    Sensitivity to algorithm parameters

One important parameter in our method of knickpoint detection is the concavity index ($\theta$). The concavity index controls the magnitude of $k_{sn}$ because it determines the values of $\chi$ (equation 6), and a higher concavity index will produce higher $k_{sn}$ values for the same channel. We ran the algorithm on Santa Cruz Island for $\theta$ values ranging from 0.05 to 0.95, in steps of 0.05.

Because the value of $\theta$ affects $k_{sn}$ order of magnitude, $\lambda$ must be adapted to keep denoising the signal. We therefore tested a wide range of $\lambda$ values for each $\theta$ value. From these tests (see Supplementary Materials 4.1) we determined default $\lambda$ values appropriate for a range of $\theta$ values. These default values are implemented internally in the code, but can be modified if needed. Sensitivity of knickpoint locations to $\theta$ using default $\lambda$ values are presented in Figure 9. This analysis shows that the general spread of the data, represented by its $z_{score}$ (difference between the data point and the mean normalised by the standard deviation), is not significantly impacted by different $\theta$ values. However, the relative magnitude of each knickpoint, measured

by changes in $k_{sn}$, depends on the chosen value of $\theta$. Therefore, if the intention of the user is to find the spatial distribution of the largest knickpoints then it is essential that $\theta$ is picked with care (see Supplementary Materials 4.2 for more illustrations of that).

Because $k_{sn}$ values are sensitive to the value of the concavity index, $\theta$, it is important to note that basins with different $\theta$ values should be analysed separately to isolate knickpoint locations. $\Delta k_{sn}$ values are therefore also dependent on the value of $\theta$ and so relative magnitudes of knickpoints and knickzones should only be compared amongst basins with the same $\theta$ value. On the other hand, the locations of knickpoints and knickzones are relatively insensitive to $\theta$ so the method can be used to determine the spatial distribution of knickpoints across large areas even in the event that the concavity index may vary spatially.

The extraction of channel steepness will also be influenced by parameters in the segment fitting algorithm (Mudd et al., 2014): the number of target nodes (noted $n_{tg}$) and the average number of nodes skipped (noted $n_{sk}$). We therefore ran sensitivity analyses on these parameters testing every combination for the following ranges of values: from 5 to 120 $n_{tg}$, and values of 1 to 4 for $n_{sk}$ parameter. Our results show that both of these parameters affect the segment lengths. Increasing either the number of $n_{tg}$ or the $n_{sk}$ parameter leads to longer segments (see Supplementary Materials 4.3 for more details). This affects the number of knickpoints detected. We also tested the number of Monte-Carlo iterations ($n_{MC}$) processed for each segment from 5 to 500, and find that the results become insensitive to $n_{MC}$ when $n_{MC} > 50$.

The results of the vertical-step knickpoint detection can change with the size of moving window that detects sudden changes in $z_{seg}$ compared to neighbouring nodes (Section 2). We tested the following combination of parameters for vertical-step knickpoint detection: $r_W$ from 10 to 200 nodes, over intervals of 10 nodes; and $T_\sigma$ from 5 to 10 over intervals of 0.5. Our results show that the extraction is insensitive to $r_W$ above a threshold minimum value, around 80 in our case. Below this value, the algorithm begins to identify steep channels as a succession of steps and will detect each node in the steep section as a knickpoint. We find that the number of extracted knickpoints becomes much higher if $T_\sigma < 6$, whereas $T_\sigma > 8$ results in very few knickpoints being detected. We therefore suggest selecting a value of $6 \leq T_\sigma \leq 8$.

The resolution of the DEM may also affect the location of extracted knickpoints and knickzones. We conducted a sensitivity analysis on raster resolution by resampling the original 1 m lidar-derived DEM into coarser grids to represent common available resolutions of 5 m (e.g., NED or NetMap), 10 m (e.g., NED or TanDEMX) and 30 m (ASTER or SRTM). Our results (see Supplementary Materials 4.7) show a decreasing number of detected knickpoints at coarser grid resolutions. This is directly linked to the amount of nodes in each river profile: as the resolution decreases, the number of nodes per river also decreases, meaning that less segments are used to extract $k_{sn}$. Therefore, less knickpoints are detected as knickpoints tend to be located near the segment boundaries. Furthermore, with lower resolution grids the knickpoints that are detected tend to represent larger-scale variations in the channel profile. Vertical-step knickpoints also tend to be identified as steepened reaches rather than purely vertical regions of the channel profile, as the grid resolution prohibits identification of small waterfalls. In order to show an overview of the algorithm performance in different field sites and DEM datasets, we extracted knickpoints from an additional test site using a 30 m DEM derived from SRTM (Supplementary Materials, Figure S21).

# 5 Discussion

## 5.1 Selecting parameter values

Ideally our method for knickpoint detection could proceed without any human supervision. Due the the method's sensitivity to grid resolution, roughness, as well as the intrinsically heterogeneous nature of landscapes, the method does however retain some user-defined parameters. The sensitivity analysis performed on the Santa Cruz Island data (Section 4.3) indicates which of these must be selected with care.

We found that changing the concavity index does not change the location of the knickpoints substantially, but it does control their relative magnitude (Section 4.3), and therefore if the user is interested in knickpoint magnitude than $\theta$ should be selected carefully (e.g., Mudd et al., 2018). Parameters linked to segmenting the $\chi$–elevation profiles (Mudd et al., 2014) that affect results are the $n_{tg}$ and $n_{sk}$ parameters (Section 4.3). Increasing both of these increases the length of the segments, where setting these parameters to smaller values result in a large number of detected changes in $k_{sn}$ which must thereafter be thinned. The one potential advantage of smaller segments is that more vertical-step knickpoints can be detected (i.e., waterfalls). Smaller segments also affect the relative values of knickpoint magnitude because short, steep reaches can be extracted and will generate high magnitude $\Delta k_{sn}$ knickpoints. If high values for the $n_{tg}$ and $n_{sk}$ parameters are used, the resulting knickpoint dataset will be sparser but will not necessarily detect local changes of $k_{sn}$ due to local layers of hard rock layer or a change in erosion process, for example. Larger segments are also less sensitive to topographic noise. After running sensitivity analyses, we recommend default parameters of $n_{tg}$ = 80 and $n_{sk}$ = 1.

Once segmentation is performed, we use the TVD routines to isolate changes in $k_{sn}$, which require an additional parameter ($\lambda$) to control the degree of denoising (equation 7). As the relative magnitude of $k_{sn}$ is controlled by the $\theta$ value, we also determine the $\lambda$ value for each value of $\theta$ that best isolates changes in $k_{sn}$ based on our sensitivity analysis (Section 4.3). However, some landscapes that are either very gentle or steep may require changes to the $\lambda$ value: low-relief landscapes may require a smaller $\lambda$ value whereas the opposite is true for steep landscapes. The user can check the efficacy of the selected $\lambda$ value by plotting $k_{sn}$ and denoised $k_{sn}$ against $\chi$ or the flow distance. Guidance on selection of $\lambda$ is described in greater detail in Supplementary Materials Section 4.1.

We also explored the possibility of using the TVD routine to denoise the river profile before extracting knickpoints in order to avoid dependency on the $\theta$ parameter. We applied the denoising routine on $\Delta elevation$ in order to reduce the amount of variation. The intensity $\lambda$ of denoising has to be manually selected and controls the amount of change from original data. Results from these tests are available in the Supplementary Materials (Figures S18-S20). We found that additional denoising is still required during the Monte Carlo segment determination of Mudd et al. (2014). We suggest that prior smoothing of river profiles needs to be carefully considered, as it unavoidably leads to some modification of the existing profile. Users of our software may, if they wish, apply a technique for denoising river profiles prior to applying our method (e.g. Schwanghart and Scherler, 2017).

The width of the combining window can also be an important factor. As explained in Section 2, segment boundaries can still be fuzzy after the denoising process, generating successions of low-magnitude slope-break knickpoints. The combining

window solves this issue by merging adjacent knickpoints within a certain radius. However, underestimating $r_{comb}$ could result in retaining some of these low-magnitude knickpoints. Overestimating its size would possibly result in shifted knickpoint locations and misrepresentation of their magnitude if unrelated knickpoints are merged. In the case where the DEM resolution is high enough to represent a close succession of knickpoints, we recommend carefully choosing a combining window smaller than the spacing between these features in order to avoid merging them.

Vertical-step knickpoint detection is controlled by two parameters: the window radius ($r_W$) and the standard deviation threshold for detecting anomalies ($T_\sigma$). Section 4.3 details the combined sensitivity analysis on these parameters and allows us to determine a set of values suitable for this analysis. However, if the user's specific aim to detect vertical-step knickpoints (assuming that the DEM precision allows it), we recommend that users precisely constrain the standard deviation coefficient, the window size and the segment size, in order to make sure that vertical-step knickpoints are extracted rather than slope-break.

Although parameters in the method may be tuned and therefore the method can be supervised, it is reproducible. Workers using the method can report on the parameter values used and others can use these to reproduce the original results. One advantage of these adjustable parameters is that users can visually inspect outputs and change parameters such that the algorithm selects "obvious" knickpoints. However, we emphasize that this is not hand picking of knickpoints: the algorithm output is a dense dataset of knickpoints. While sorting the dataset, once a threshold or statistical criteria is selected, all knickpoints and knickzones matching the selection are chosen. This means that one cannot eliminate knickpoints that qualitatively appear to be in the "wrong" place. As highlighted in Figure 7, human selected knickpoints and knickzones frequently produce biased knickpoint datasets that both include and exclude knickpoints and knickzones that have the same magnitude. We note that because the segmentation algorithm uses a Monte Carlo sampling routine (Mudd et al., 2014) there may be minor differences in results between two users, but by using a reasonable $n_{MC}$ (>50) the results from one run to the next are nearly identical.

## 5.2 Quantification and selection of knickpoints

The aim of extracting knickpoints is mainly to link knickpoint location and magnitude to a specific event resulting in landscape transience (e.g., Crosby and Whipple, 2006). Therefore, an important step is to isolate the most significant knickpoint features from the dense raw dataset in order to interpret landscape evolution, which can be done using knickpoint magnitude. Knickpoint magnitude may be affected by the calculation of $k_{sn}$ using the gradient of segments in $\chi$–elevation space. Depending on the relief, and particularly with a high value of $\theta$, the absolute values of $\chi$ coordinates and associated elevation can differ by an order of magnitude. If the values of $\chi$ are low compared to the values for elevation, any changes in elevation at a knickpoint will result in a much higher segment gradient than if the $\chi$ values are of a similar magnitude as the elevation. This can result in the exaggeration of knickpoint magnitude in high relief landscapes, for example, where it is more likely that $\chi$ values will be lower than the elevation values and eventually results in a bias during the sorting. We therefore suggest that, in such cases, $A_0$ from equation 3 should be set such that the value of the $\chi$ coordinate is the same order of magnitude as the elevation. However, if $A_0 \neq 1$, then the gradient of the segment corresponds to $M_\chi$ in equation 6, rather than to $k_{sn}$. We wish to emphasise that this does not change the relative ordering between knickpoints. We illustrate this relationship by running a simple sensitivity analysis on the Santa Cruz Island dataset, with a range of $A_0$ varying from 1 to 500 (Figure 10). This sensitivity analysis shows

that, as $A_0$ is increased, the extreme values of $\Delta k_{sn}$ within the dataset are reduced, so that the effect of low absolute $\chi$ values on the gradient calculation is diminished. As for $\theta$ (see section 4.3), knickpoint absolute magnitude (*i.e.,* the direct value of $\Delta k_{sn}$ and $\Delta z_{seg}$) cannot be compared if calculated with different $A_0$ from equation 3. However the location of the isolated main knickpoints can still be compared.

Our sensitivity analyses suggest that two different approaches may be used to select knickpoints. The first of these is that a single $\theta$ and $A_0$ can be fixed for an entire landscape: the knickpoint magnitudes can directly be used to isolate the main knickpoint locations and relative importance. However, this approach may lead to some errors due to inevitable landscape heterogeneity over larger scales. The second approach is to calculate $\theta$ and $A_0$ values separately for individual basins, which allows knickpoints to be extracted with greater precision than if a single value is set for the entire landscape. However, this approach means that the knickpoint extraction has to be processed independently for each catchment, and only the location (e.g., latitude, longitude, elevation) are comparable between different catchments. Which approach is taken is dependent on the aims of each particular study, and should be carefully considered on a case-by-case basis.

## 5.3   Knickpoint and knickzone morphology

Along with the calculation of knickpoint magnitude, our algorithm allows the characterisation of knickpoint morphology. We can identify different knickpoint or knickzone types by i) identifying locations where $k_{sn}$ increases downstream (positive slope break knickpoints); or ii) identifying locations where $k_{sn}$ decreases (negative slope break knickpoints); and iii) identifying locations where a sudden change in elevation occurs (vertical step knickpoints). This approach is suitable to identify the most common morphologies described in the literature (e.g Haviv et al., 2010; Kirby and Whipple, 2012). However, we wish to emphasise that this algorithm can also be used to focus on one particular knickpoint morphology. For example, the classical convex-upwards knickpoint expression (e.g., Knopf, 1924) can be isolated by only displaying the knickpoints with a drop of $\Delta k_{sn}$ (Figure 11b). In order to examine steepened reaches or knickzones, we can also isolate locations where $\Delta k_{sn}$ increases. Finally, waterfall detection can be achieved, if the resolution of the DEM allows it, by focusing on locations with a jump in $z_{seg}$. We provide all these different knickpoint types for the Smugglers Catchment in the Supplementary Materials (Figure S12).

## 5.4   Comparison with other knickpoint extraction techniques

For each of our two study sites, we have presented performance metrics of our method compared to knickpoints selected by humans. We find that our method has a high sensitivity, meaning that nearly all human-identified knickpoints were selected by the algorithm, but a lower reliability. This suggests that our algorithm also identifies many changes in channel steepness which are not selected as knickpoints through field mapping techniques. This raises the question of whether algorithmic selection of knickpoints is more or less trustworthy than those selected by humans.

Knickpoints identified for geomorphic studies should be reproducible, in that two workers should be able to select the same locations and magnitudes from the same river profile. This is challenging when mapping features in the field, as different workers may have different criteria for what constitutes a knickpoint. Furthermore, knickpoint selection should be objective:

the same morphological criteria should be used to identify all features in the dataset. A common problem with field mapping by humans is that some specific features are picked in order to interpret a signal, whereas others with a similar morphology may be omitted. Our approach allows the production of an objective dataset of knickpoint locations and magnitudes that can be later correlated by the user with process-based interpretations. Algorithmic extraction also allows coverage of much larger areas compared to field mapping, that can later be calibrated with additional data (e.g., Crosby and Whipple, 2006). As illustrated by our accuracy metrics, our algorithm produces dataset significantly denser than hand picked knickpoints. However it is possible to thin the number of knickpoints by applying thresholds metric values selected based on statistical criteria, and making the number of identified features similar to human-picked datasets. Such a process is objective in the sense that no hand selection is involved: only the morphology drives the thinning.

To provide a full assessment of our methods, we compare its output to the one generated two other algorithms as explained in section 1.1.3: TopoToolbox (Schwanghart and Scherler, 2014); and KZ-picker (Neely et al., 2017). Figure 11a expresses the differences between KZ-picker and our algorithm for a single channel, where KZ-picker identifies the main knickzone (in red) and quantifies its magnitude by the difference in elevation between the toe and lip of the knickzone. The purpose of the KZ-picker is to find broad zones of steepened channels and is less granular than our method (e.g., Section 4.1). It is also not constructed to identify discrete vertical-step knickpoints. Because the raw output from our algorithm is however denser than the KZ-picker, main knickpoints from our algorithm require more sorting based on their magnitudes which results in extra steps to explore the data.

Figure 11b provides a basin-wide comparison of our outputs with those from TopoToolbox (Schwanghart and Scherler, 2014), with a tolerance parameter of the TopoToolbox method fixed to 5. In order to ensure that the comparison is valid we only compare it to our negative $\Delta k_{sn}$ knickpoints, which should quantify similar features. The TopoToolbox method effectively identifies the main knickpoints expressed by the difference to an idealised profile that is concave-up. However, reducing the tolerance parameter increases the number of knickpoints detected (e.g., 10: 12 knickpoints, 5: 44 knickpoints, 1: 343 and 0.1: 2234) meaning that the TopoToolbox method can result in a network of knickpoints that has a similar density to our method. However the TopoToolbox method relies on profiles in elevation plotted against flow distance and so further processing is required to analyse changes in channel steepness using this method. Because selection of knickpoints in this method is not normalised for drainage area, the largest knickpoints selected may not correspond to the largest changes in channel steepness. However it has fewer parameters and is more computationally efficient than our method.

While the KZ-picker and the TopoToolbox methods are well adapted for identifying specific types of knickpoint, neither allows the separate identification and quantification of positive slope-break, negative slope-break, and vertical-step knickpoints. Each method produces slightly different data products that can be used to interpret different components of the channel network, making these methods complementary.

Finally, we chose to build our change point detection method using the TVD routine (Condat, 2013). However, as explained in Section 2, alternative methods could be used. The algorithm therefore provides the raw data before the TVD routine, meaning that this data can be ingested by other change point detection techniques, *e.g.,* the methods reviewed in Truong et al. (2018) and its associated open-source code.

# 6 Conclusions

We have developed a new method for extracting knickpoints and knickzones from topographic data. Our method extracts slope-break knickpoint locations using changes in channel steepness $k_{sn}$, calculated by combining a statistical method for segmenting channels into reaches of different channel steepness (Mudd et al., 2014) and a recently introduced denoising technique (Condat, 2013). The method also identifies vertical-step knickpoints by searching for breaks in elevation between channel segments of similar channel steepness. Our algorithms provide a dense dataset of objectively extracted knickpoint locations, along with the relative magnitude of each knickpoint defined by either the change in channel steepness (for slope-break knickpoints) or the jump in elevation (for vertical-step knickpoints) to quantify knickpoints morphologies.

We tested our algorithm on two datasets where knickpoints were independently field mapped, and found that our method successfully extracted the human-identified knickpoints in the vast majority of cases. In general the method identifies more knickpoints compared to field mapping, as illustrated by our accuracy metrics, especially in the case of knickzones where one broad steepened reach may result in multiple discrete segments in $\chi$-elevation space. We provide tools for sorting and thinning the dense dataset in order to isolate the most significant breaks in the channel profile without involving any human-based selection. Resulting knickpoints can be compared with lithological, climatic, or tectonic datasets. Our method therefore provides an objective, systematic and reproducible technique for quantifying knickpoints and knickzones, which can then be used to inform process-based interpretations of landscape evolution.

*Code and data availability.* Code used for analysis is located in the LSDTopoTools github repository: https://github.com/LSDtopotools/LSDTopoTools_ChiMudd2014, and scripts for visualising the results can be found at https://github.com/LSDtopotools/LSDMappingTools. We have also provided documentation detailing how to install and run the software which can be found at https://lsdtopotools.github.io/LSDTT_documentation. As part of the supplementary information we have also provided example parameter files which can be used to reproduce the results of all analyses performed in this study.

*Author contributions.* BG designed the study with contributions from other authors. BG designed the algorithms and wrote the code with contributions from SMM and FJC. BG and DP ran the analysis on test sites. BG wrote the paper with contributions from other authors.

*Competing interests.* We declare no competing interests.

*Acknowledgements.* We thank Emma Graf for testing the software. BG was funded by European Union Initial Training Grant 674899 – SUBITOP. SMM was supported by NERC grant NE/J009970/1, FJC was supported by a Geo.X fellowship and DP had support from the Coordination for the Improvement of Higher Education Personnel (CAPES) under a Science without Borders fellowship BEX 12000/13-2.

We thank the German Aerospace Center (DLR) for granting access to TanDEM-X data as part of the project DEM_GEOL1345. We would like to thanks Giulia Sofia, Wolgang Schwanghart, Stefan Hergarten and an anymous reviewer for their helpful comments and suggestions.

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

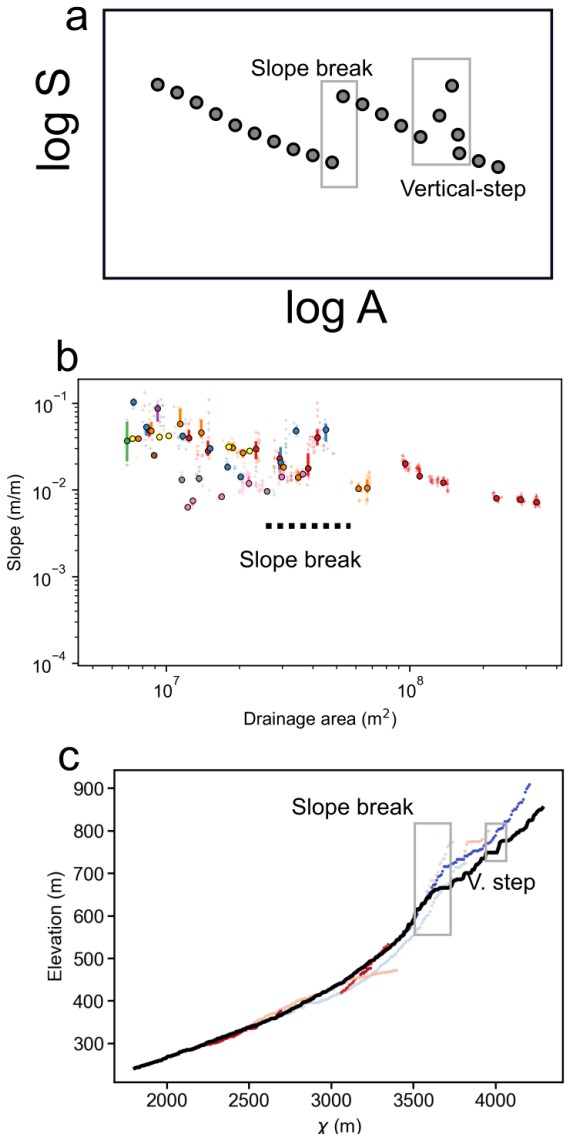

**Figure 1.** Different methods to detect knickpoints. (a) Cartoon showing how vertical-step and slope-break knickpoints appear in slope–area plots, adapted from Kirby and Whipple (2012). (b) A slope–area plot derived from SRTM 30 metres resolution data in Romania; the catchment's outlet coordinates are 45.252842, 26.375697 (WGS84). Different colours represent different tributaries, small '+' symbols are individual data points and circles are logarithmically binned data. A single slope-break knickpoint can be interpreted but minor knickpoints are more difficult to extract. (c) The same basin represented in a $\chi$–elevation plot, using $\theta = 0.15$.

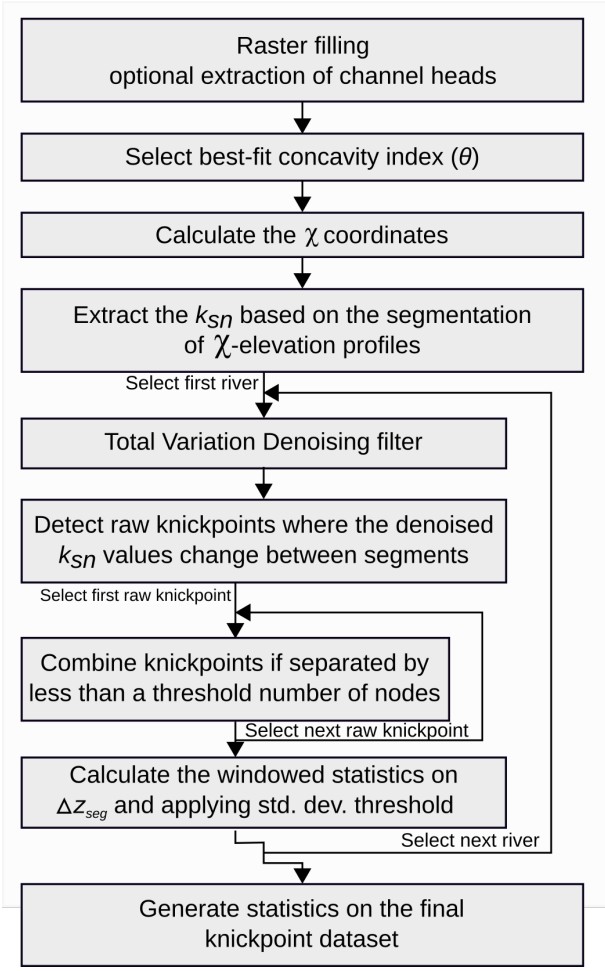

**Figure 2.** Flowchart of the knickpoint detection algorithm.

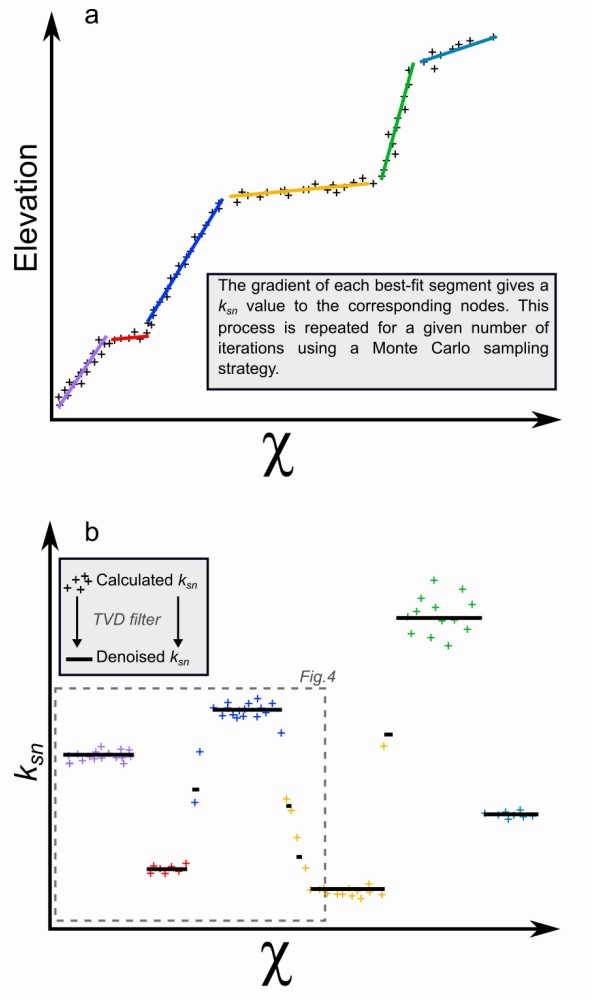

**Figure 3.** Extraction of normalised channel steepness ($k_{sn}$) from a river profile. (a) Example of best-fit segmentation (Mudd et al., 2014) where '+' symbols are individual data points and the coloured lines are the segments. (b) The associated plot of $k_{sn}$ plotted as a function of $\chi$ coordinate. The segmentation output results in some noise due to iterative sampling of the channel network ('+' symbols). Total Variation Denoising filter (Condat, 2013) is then applied on the signal to extract the main variations in $k_{sn}$.

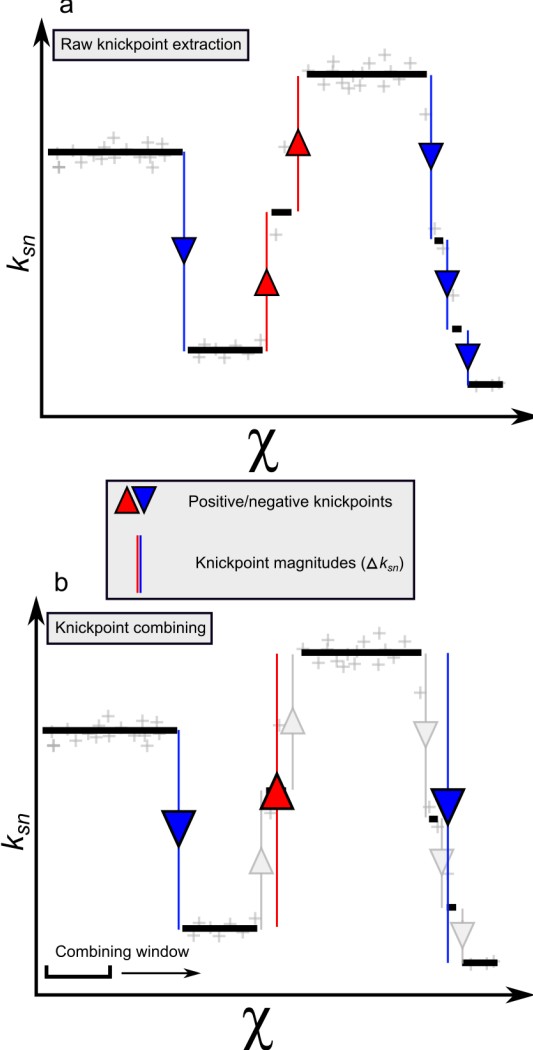

**Figure 4.** Knickpoint extraction from the denoised $k_{sn}$ profiles. (a) The first step extracts all variations of $k_{sn}$, quantifying each with $\Delta k_{sn}$, which we call the "raw" knickpoint dataset. Negative and positive changes represents decreases or increases of $k_{sn}$, respectively. (b) After detection of changes in $k_{sn}$, knickpoints are combined. All knickpoints within a node window will be combined, summing their values (i.e., a sum of $\Delta k_{sn}$. This process is repeated as long as the subsequent raw knickpoint is within a node window and as long as the polarity (*i.e.*, if it is negative or positive) does not change.

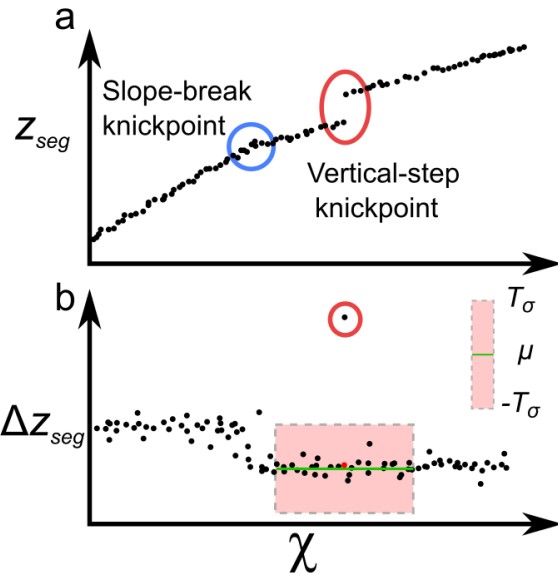

**Figure 5.** Extraction of knickpoints from the segmented elevation (equation 5). (a) Expression of a vertical-step knickpoint in a $\chi - z_{seg}$ profile compared to a slope-break knickpoint. (b) Representation of the identification window and the corresponding standard deviation around the reference node (in red). $\mu$ is the mean and $T_\sigma$ the coefficient applied to the standard deviation. This process is repeated for each node. Reference nodes outside their own window are considered to be outliers.

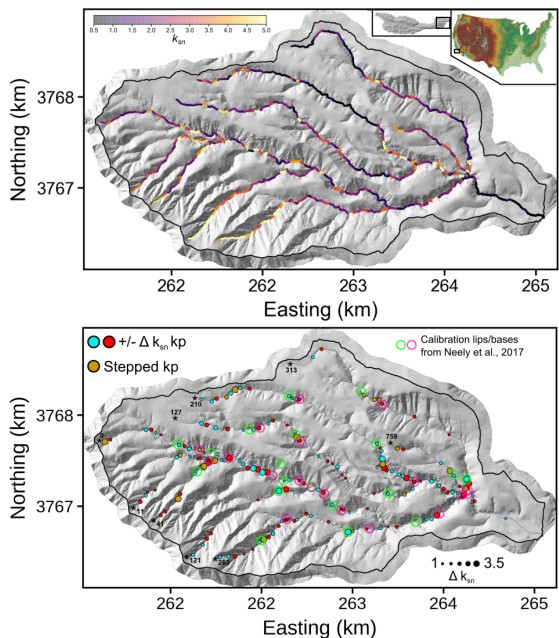

**Figure 6.** The test location on Santa Cruz Island, CA, USA. (a) Map of channel network extracted with the Pelletier method (Pelletier, 2013), and is coloured by $k_{sn}$ value calculated with Mudd et al. (2014). (b) Extracted knickpoints plotted after thinning the dataset as described in Section 4.1. The purple and green circles respectively represent the calibration knickzones' bases and lips with the 50m radius used for assessing algorithm performances. Stars and associated numbers are source numbers, which can be compared to Figure 7. Topographic data is 1 meter precision lidar DEM (see Supplementary Materials 1 for metadata), reprojected in WGS84 UTM zone 11N.

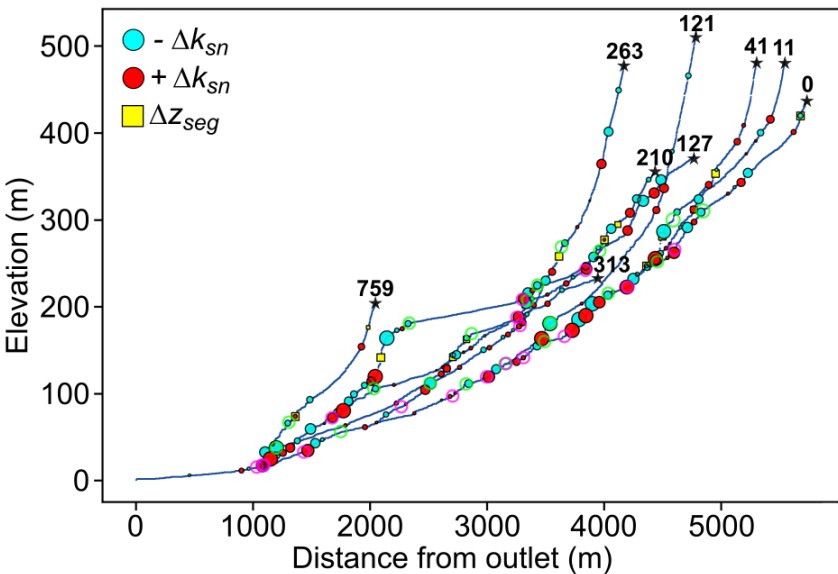

**Figure 7.** Knickpoints extraction for Santa Cruz Island, CA, USA shown for the channel long profiles. These are the same knickpoints depicted in Figure 6b. The stars and associated numbers correspond to the source numbers, and green and mauve circles correspond to the lips and bases of mapped knickpoints from Neely et al. (2017).

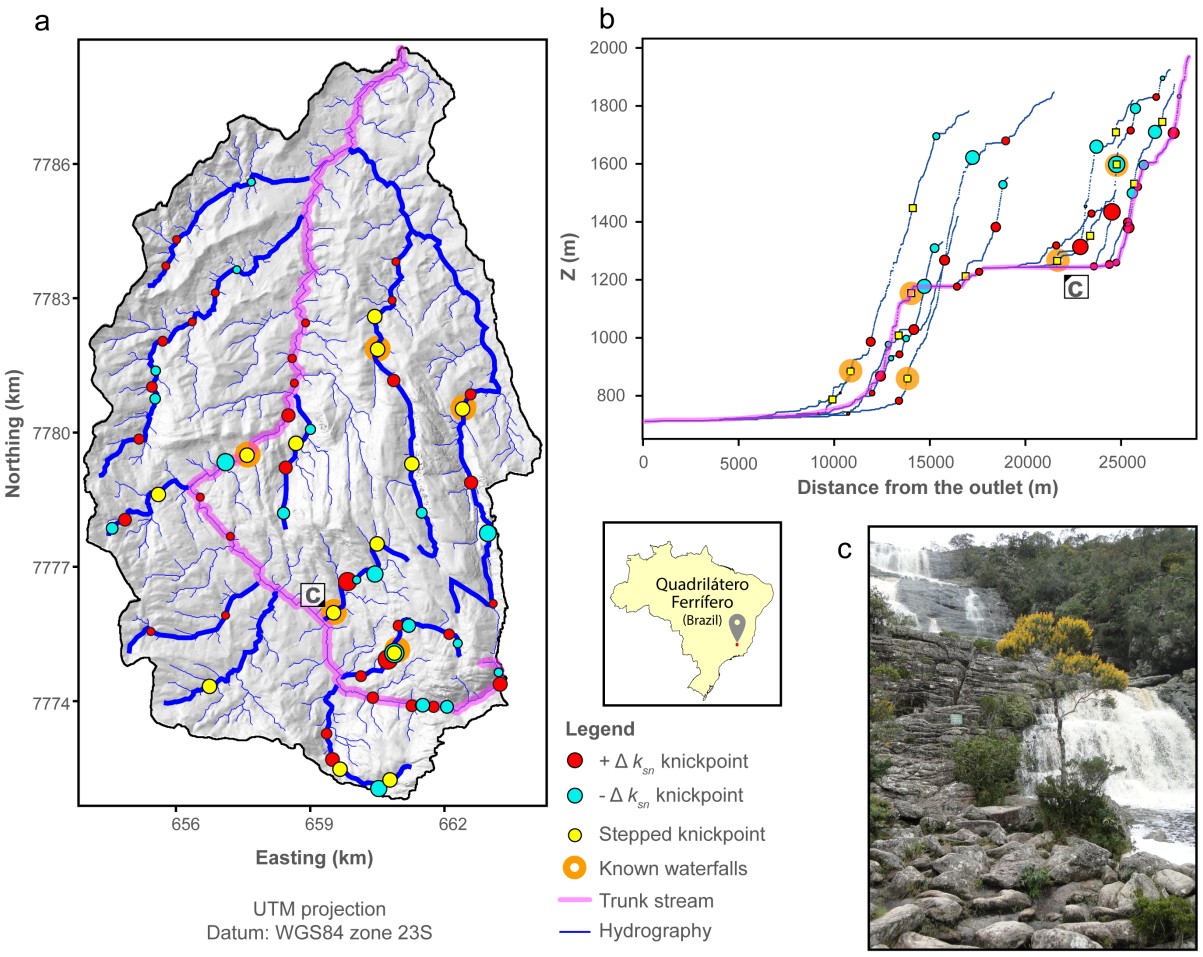

**Figure 8.** Knickpoint extraction on the Ribeirão Caraça basin (Caraça Range, QF, Brazil). (A) Map of knickpoints extracted with the algorithm after thinning the dataset as described in Section 4.2. Most of the calibration knickpoints are expressed by a succession of knickpoints detailing along-channel increases/decreases in $k_{sn}$. Streams depicted in B are shown as thick blue lines. (B) Longitudinal profile of the trunk stream (the Ribeirão Caraça river) highlighting the performance of the algorithm in picking along-channel breaks in steepness. (C) Example of known waterfall (*i.e.*, waterfall with a name) in the field; in this case, the Cascatinha waterfall. This waterfall features an elevation break of 40 m. Other known waterfalls include the Cascatona, Bocaina, Brumadinho, and Quebra-ossos waterfalls.

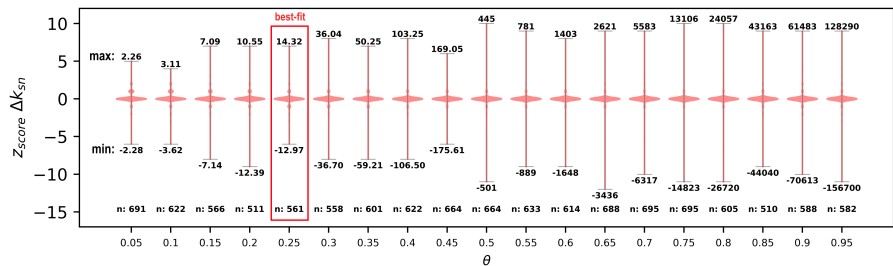

**Figure 9.** Sensitivity of the knickpoint extraction to the concavity index ($\theta$). As different values of $\theta$ result in different values of $k_{sn}$, we use a normalised $z_{score}$ (*i.e.* the difference to the mean normalised by the standard deviation) to compare the overall spread of $\Delta k_{sn}$. The plot shows probability distributions of the $z_{score}$ of $\Delta k_{sn}$ represented by violin plots calculated with a Kernel Density Estimation (bandwith = 0.20). The outliers and their relative magnitudes are affected by this parameter, whereas the general data distribution remains similar. The 'min' and 'max' stated above and below the violin plots respectively represents the minimum and maximum $\Delta M_\chi$ for each run.

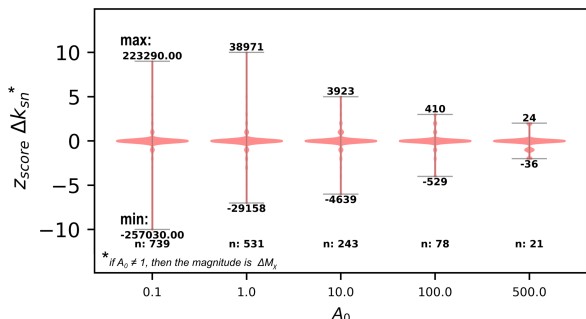

**Figure 10.** The effect of varying $A_0$ on knickpoint extraction (equation 3). The reference area ($A_0$) will affect knickpoint magnitude and can be increased to reduce exaggerations in $\chi$-elevation gradients. Changing $A_0$ does not affect the relative order of knickpoints: the largest knickpoints remains the largest for all values of $A_0$. Increasing $A_0$, however, reduces the spread in the $z_{score}$ of the changes in channel steepness. This value has to be set only if necessary (e.g., if the high-gradient effect is important): $A_0 \neq 1$ implies that the magnitude is not $\Delta k_{sn}$ but $\Delta M_\chi$ from equation 6. Moreover, overestimating $A_0$ can mask knickpoints that would be detected with $A_0 = 1$ m$^2$. The 'min' and 'max' stated above and below the violin plots represent the minimum and maximum $\Delta M_\chi$ for each run.

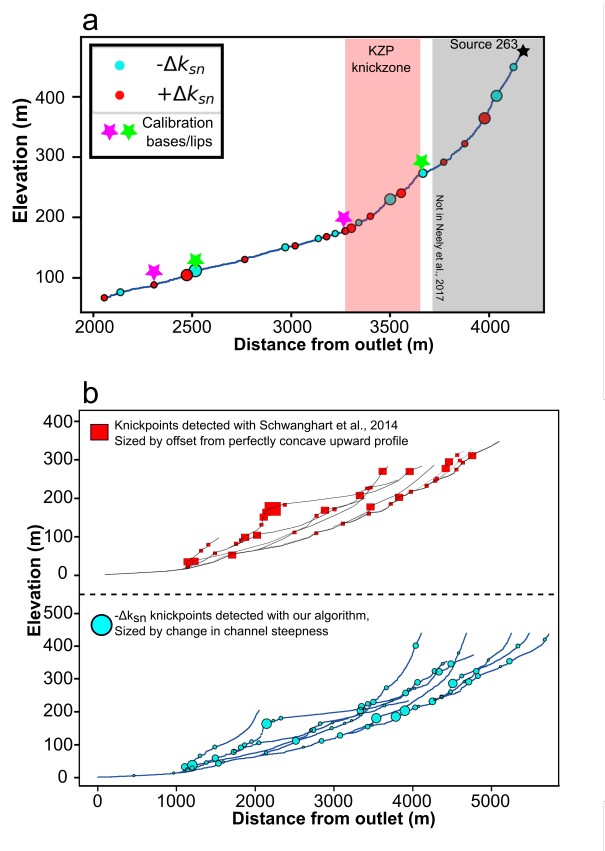

**Figure 11.** Comparison of results on the Smugglers catchment from our algorithm and the most recent similar ones. (a) Results for a single source from KZPicker (Neely et al., 2017) and our results. The results from Neely et al. (2017) are directly taken from their study to ensure objectivity. Only the slope-break knickpoints are displayed to make the comparison valid. (b) Basin-wide comparison between our algorithm outputs and the one recently implemented in Schwanghart and Scherler (2014) using tolerance = 5. We only display the knickpoints showing a decrease of $k_{sn}$, in order to provide a relevant comparison with the knickpoints morphology detected by Schwanghart and Scherler (2014). Differences in channel length are due to different methods for extracting channel heads between the two techniques.

**Table 1.** Parameter values used for the two field sites. Differences in parameter values between the two sites is due to differing DEM resolution (1 metre for Santa Cruz Island, and 12 metres for the Ribeirão Caraça). Sensitivity to these parameters is described in Section 4.3. Note that although the parameter values have been carefully optimized for knickpoint analysis, we suggest the below values as defaults for each of these two data resolutions in order to allow a rapid initial knickpoint extraction for other landscapes.

| Parameter name | Santa Cruz Island, USA | Ribeirão Caraça, Brazil |
|:---:|:---:|:---:|
| $n_{tg}$ | 30 | 50 |
| $n_{sk}$ | 1 | 1 |
| $n_{MC}$ | 100 | 100 |
| $\lambda$ | 1.7 | 0.3 |
| $r_{comb}$ | 10 | 30 |
| $T_{\sigma}$ | 7 | 7 |
| $r_W$ | 120 | 100 |

**Table 2.** Accuracy metrics for calibration site I (Smugglers Catchment, California, USA)

| Source key | TP | FP | FN | Total detected |
|:----------:|:--:|:--:|:--:|:--------------:|
| 0 | 26 | 15 | 4 | 41 |
| 11 | 0 | 15 | 0 | 15 |
| 41 | 4 | 5 | 0 | 9 |
| 121 | 2 | 5 | 1 | 7 |
| 127 | 17 | 15 | 1 | 32 |
| 210 | 17 | 9 | 0 | 26 |
| 263 | 11 | 13 | 0 | 24 |
| 313 | 10 | 4 | 1 | 14 |
| 759 | 4 | 5 | 0 | 9 |
| Total | 91 | 81 | 7 | 177 |

$s = 0.93$, $r = 0.53$ and $q = 0.51$

**Table 3.** Accuracy metrics for calibration site II (Ribeirão Caraça basin, Caraça Range, QF, Brazil)

| Source key | TP | FP | FN | Total detected |
|---|---|---|---|---|
| 0 | 17 | 13 | 2 | 32 |
| 1 | 6 | 5 | 1 | 12 |
| 5 | 9 | 1 | 0 | 10 |
| 22 | 4 | 2 | 0 | 6 |
| 37 | 3 | 2 | 1 | 6 |
| 56 | 4 | 2 | 1 | 8 |
| 88 | 9 | 7 | 1 | 17 |
| 114 | 5 | 2 | 1 | 9 |
| 139 | 8 | 5 | 0 | 14 |
| 151 | 4 | 4 | 1 | 10 |
| 252 | 6 | 8 | 0 | 15 |
| Total | 75 | 51 | 8 | 139 |

$s = 0.89$, $r = 0.60$ and $q = 0.56$