# Peer review of "A segmentation approach for the reproducible extraction and quantification of knickpoints from river long profiles"

_Earth Surface Dynamics, 2018_

## Referee Comment (RC1) · S. Hergarten (Referee) · 16 Oct 2018

The authors present a technique for finding and quantifying knickpoints in river profiles. The group of coauthors belongs to the forefront researches in the field of developing morphometric tools having already developed a large software suite.

The manuscript is quite well written and reviews previous approaches properly. But despite this overall positive impression I am not convinced that this undoubtedly nice piece of work should be a full research paper in Earth Surface Dynamics.

First, I am not completely convinced that the automatic detection of distinct knickpoints

is still such a great step in fluvial geomorphology. Knickpoints are fundamental for understanding the effect of sudden temporal changes or discontinuities in lithology, and they were a primary measure in morphometry at times where high-resolution DEMs were not widely available. However, one may question whether finding such distinct points automatically has really a greater potential than analyzing river profiles or even entire drainage networks as a whole. This might reduce the importance of this work a, but does of course not question the merit of this work.

But as my most severe doubt, I see the new aspects presented here as a piece of a mosaic. If I understood the concept correctly, the new part is applying the TVD method from signal processing to the $k_{sn}$ values described in Sect. 2.3.1., while the earlier steps of the analysis are apparently based on previous work. And this key point is not explained very well. I would have expected more explanation why this is a particularly good concept in the context of river profiles going beyond the comparison of the entire procedure with other approaches.

Taking into account that entire packages such as TopoToolbox are published as a short communication in Earth Surface Dynamics, the recent manuscript would not be well-placed as a full research paper in my opinion. In order not to be misunderstood – this is a nice piece of work, but if we are honest, each comprehensive package such as LSDTopoTools from your group contains many important and innovative components, and it would not be realistic to derive a full research paper from each of them.

My recommendation would be either focusing the manuscript on the essential new part and submit it as a (very) short communication or including the methodical aspects into a later paper where scientific results are derived using the method going beyond the test cases presented here.

No matter in which direction a revised version goes, there are a few minor points that deserve some more attention. I am, e.g., not sure whether the definition of $\chi$ was indeed introduced in the conference contribution by Royden et al. (2000) more than

**ESurfD**

Interactive
comment

10 years before it became popular; at least I did not find it in the cited abstract. As a second example, the lower sections of page 5 read as if 2014 was more recent than 2017. However, these minor points can easily be fixed.

---

## Referee Comment (RC2) · W. Schwanghart (Referee) · 23 Oct 2018

Gailleton et al. present a method that automatically extracts knickpoints from longitudinal river profiles. The algorithms developed by the authors are well described and are implemented in LSD TopoTools, a terrain analysis software written and maintained by the authors. The algorithms are tested against hand-picked knickpoints and those derived with other software, and the code is publicly available. Overall, the manuscript is very well written and nicely illustrated. I have no concern about this paper being appropriate for the journal ESURF. To this end, I only have a few questions and some specific comments.

[Figure]

- Would it make a difference, if you first smooth the elevation values using the TVD-approach and then calculate ksn? The smoothness-parameter would then be independent of theta.

- Detecting knickpoints by identifying gradient-changes of ksn could also be achieved by calculating the profile curvature of the elevation data in chi-space. Similarly to $M_\chi$, this could be $C_\chi$ (or something similar). Of course, mathematically, this is the same. In addition, curvature is strongly affected by noise in the river long profile. However, using curvature instead of gradients of gradients is slightly more elegant and smoothing curvature might directly yield the peaks and troughs that you are looking for.

- Detecting changepoints in noisy data is a common topic in signal processing and statistics (see e.g. Truong et al., 2018). I wonder whether some of the techniques of knickpoint identification could actually be applied in a more formal statistical framework.

In conclusion, I think that the paper needs only minor revisions.

**Specific comments**

6, 25: Filling might cause problems, because it can generate some large steps. Carving might be a better alternative.

8, 12: How much does "combining knickpoints" (2.3.2) actually affect the objective to identify the precise location of transitions between segments? It seems to me that knickpoint merging will let you pick knickzones, rather than knickpoints.

Eq. 7: Denoising: The TVD algorithm (Eq. 7) is similar to the smoothing approach by Schwanghart and Scherler (2017), with the difference being the applied smoothness penalty. It would be interesting to know why you chose a gradient penalty instead of a curvature penalty. Wouldn't the gradient penalty require the horizontal distance in

the denominator as the node-to-node distance may change depending on whether the node is a cardinal or diagonal neighbor?

12, 20: I was wondering about this error radius when reading through section 2.4. Consider to mention the radius also there. Did you use the same radius in the Brazilian test case?

References:

Schwanghart, W. and Scherler, D.: Bumps in river profiles: uncertainty assessment and smoothing using quantile regression techniques, Earth Surface Dynamics, 5(4), 821–839, doi:10.5194/esurf-5-821-2017, 2017.

Truong, C., Oudre, L. and Vayatis, N.: A review of change point detection methods, [online] Available from: https://arxiv.org/abs/1801.00718 (Accessed 23 October 2018), 2018.

---

## Referee Comment (RC3) · Anonymous Referee #3 · 13 Nov 2018

This work presents an improved approach for the extraction and quantification of knickpoints from river long profiles. The work is well written and clear in its goals.

Having said that, in my opinion, there are two major issues to fix before getting it published: (1) the format of the publication that at my eyes is a technical note, not a full paper; (2) representativeness of study area considered in this study and its application in complex geomorphology landscapes.

(1) In the hands of the Esurf editor the decision, but at my eyes, this paper, mainly focused on an improved method, among others available in the literature, should be structured as a technical note.

[Figure]

(2) I'm not sure, but are the study areas presented, enough to guarantee a robust analysis of the capability of the given method to work objectively in different landscape contexts, and complex morphological conditions? The impression (but maybe I'm wrong) is that the landscape morphology of those areas is quite gentle. . .

For other points, I'm generally in line with the feedback provided by reviewer #1.

---

## Author Response (AR1)

THE UNIVERSITY *of* EDINBURGH
**School of Geosciences**

Boris Gailleton
*School of Geosciences*
*University of Edinburgh*
*Drummond Street*
*Edinburgh, EH8 9XP*

*Email: b.gailleton@sms.ed.ac.uk*

Giulia Sofia
Associate Editor, Earth Surface Dynamics

December 10, 2018

Dear Dr. Sofia,

Thank you for considering our manuscript 'A segmentation approach for the reproducible extraction and quantification of knickpoints from river long profiles' for publication in ESURF. We also thank the reviewers for the constructive feedback we received that helped us to improve the quality of the manuscript. Overall, all the reviewers acknowledged the quality of the work, and raised minor technical concerns. Reviewers 1 and 3 were however not convinced of the format of our submitted work and questioned its relevance as a full research paper for ESURF. In our response we have replied to each of the comments, and outlined why we believe that the format of our contribution is suitable for publication.

We have edited the manuscript, the supplemental materials and the open-source code to reply to the comments. Please find below the response letters to each of the reviewers, with the specific changes made to the manuscript associated to each comment. Original comments from reviewer are displayed in italics and our response is in normal font.

The main changes we have made to the manuscript are: (i) integrating our approach in a more statistical framework with relevant vocabulary and references; (ii) adding discussion and methods about Digital Elevation Model preprocessing (e.g., carving DEMs as an alternative to filling, or correcting bumps on river profiles). We have also clarified points that reviewer queried throughout the manuscript which are also detailed in the response letter. We hope these changes and clarifications have significantly improved the manuscript and we wish to thank the reviewers again for the constructive suggestions.

Sincerely,

Boris Gailleton

**Reviewer 1: Stefan Hergarten**

We thank reviewer 1 (Stefan Hergarten) for his thoughtful review on our manuscript. Overall, the reviewer is not convinced by the merit of producing a full research paper out of developing a reproducible method to quantify knickpoint morphology from river long profiles. We disagree: we explain why we strongly believe that our manuscript would be of benefit to the readers of ESURF and the wider geomorphic community as a full research paper below. In addition we clarify a number of points raised by the reviewer.

*First, I am not completely convinced that the automatic detection of distinct knickpoints is still such a great step in fluvial geomorphology. Knickpoints are fundamental for understanding the effect of sudden temporal changes or discontinuities in lithology, and they were a primary measure in morphometry at times where high-resolution DEMs were not widely available. However, one may question whether finding such distinct points automatically has really a greater potential than analyzing river profiles or even entire drainage networks as a whole. This might reduce the importance of this work a, but does of course not question the merit of this work.*

We acknowledge that this manuscript describes a method and neither explores geomorphic processes nor presents a detailed case study. The reviewer points out that high-resolution DEMs are becoming more widely available, and in addition knickpoints are fundamental features to help us understand landscapes. Over the last two decades dozens of papers have been published making inferences about how channels incise, and how landscapes evolve and have evolved though time on the basis of the locations of knickpoints and knickzones.

The reviewer then suggests that there is greater potential for "analyzing river profiles or even entire networks as a whole". Analyzing river profiles and networks how? Any quantitative analysis of profiles or an entire network needs a method. Choices made in creating the method have implications for the results, and therefore the interpretation of the results. So we strongly believe that these choices and their implications should be clear to workers using the method. Which is why we think methods papers are important.

We do not understand the reviewer's contention that knickpoints are "fundamental for understanding the effect of sudden temporal changes or discontinuities in lithology", but at the same time not worth finding. The numerous studies which use the spatial location, magnitude, and evolution of knickpoints suggests that understanding discontinuities in river steepness is important, and that their spatial distribution reveals important information about landscape evolution. They will exist in specific locations regardless of the resolution of the topographic data. We have tested the algorithm using lidar data and our sensitivity analysis with regards to resolution degradation suggests that knickpoints found in 1 metre resolution data can also be found in lower resolution data. That is because they are distinct topographic entities with a distinct location. So we reject the suggestion that high resolution data will make knickpoint detection obsolete. Flying lidar over a landscape doesn't make the waterfalls disappear.

Following on from this, we set out briefly here why we believe our study is useful to the community. Techniques for identifying knickpoint locations have a number of pitfalls, and choices made by authors can alter the results: for example, selection of knickpoints by humans is extremely difficult to reproduce. The main motivation for our approach is reproducibility and not pure automation, which is a more minor point of this study. Interpretation of knickpoint locations strongly depends on method choice. In this contribution we have explored the implications of these choices, and compared them with methods of other authors as well as independent field datasets. This allows future workers to make informed decisions about different knickpoint extraction methods and know the strengths and weaknesses of these

methods, all of which will affect the confidence and nature of geomorphic interpretation of the results.

*But as my most severe doubt, I see the new aspects presented here as a piece of a mosaic. If I understood the concept correctly, the new part is applying the TVD method from signal processing to the ksn values described in Sect. 2.3.1., while the earlier steps of the analysis are apparently based on previous work. And this key point is not explained very well. I would have expected more explanation why this is a particularly good concept in the context of river profiles going beyond the comparison of the entire procedure with other approaches.*

As is the case with the development of many methodologies, ours relies on many previous studies. The most important is an algorithm from Mudd et al. (2014) that proposes a statistical framework to derive $\chi$-elevation gradient ($k_{sn}$ in our case) using a segmentation approach. We set out in the manuscript (i) the reasons this approach is relevant to build upon; (ii) the difficulties of objectively identifying knickpoint location only using its raw results; and therefore (iii) the motivation of our method development. Adapting a TVD algorithm is the first step that focus the dataset on discrete significant gradient change (e.g., slope-break knickpoints). After adapting this algorithm we develop an entire process described in the manuscript to extract and transform these discrete changes in $k_{sn}$ into dataset of objective and quantified knickpoints. Furthermore, we propose an additional new approach for vertical step knickpoints. Finally, we describe strategies to interpret and thin this dataset while keeping full reproducibility of results for two fairly different case studies, to avoid constraining the algorithm to one single case study and context. The effect of parameters and datasets (e.g., grid resolution) are explored and discussed with extended sensitivity analysis in the specific context of knickpoint extraction. We believe that all these additions make our method significantly different from the other algorithms it is based on and justify the relevance of our manuscript.

*Taking into account that entire packages such as TopoToolbox are published as a short communication in Earth Surface Dynamics, the recent manuscript would not be well placed as a full research paper in my opinion. In order not to be misunderstood  this is a nice piece of work, but if we are honest, each comprehensive package such as LSDTopoTools from your group contains many important and innovative components, and it would not be realistic to derive a full research paper from each of them. My recommendation would be either focusing the manuscript on the essential new part and submit it as a (very) short communication or including the methodical aspects into a later paper where scientific results are derived using the method going beyond the test cases presented here.*

Providing open-source and documented code is crucial for making our research easily reproducible, testable and improvable. However it also generates a significant risk of mis/over interpretations of its results, as the software would ultimately produce results in any context. We therefore believe that it is crucial to provide such analysis with extended discussion on its use and in comparison to other existing methods, sensitivity analysis on the different parameters, example of uses on different landscapes, and clear statements on how to constrain it. Other scientists who might come to use our methods should be aware of what the algorithms can provide and what the limitations might be. The reviewer suggests that we drop this analysis in the appendix of a case study. However, inadequate discussion of methodology along with failure to publish software is one of our major frustrations. We do not agree with the suggestion that we follow this approach. The reviewer then suggests we just publish an overview of the general software. This again we feel would be a major disservice to users of our methods since such a paper could never go into the details of each method; the development and testing of each method typically represents many months of effort, not to mention the many years of CPU time we devote to testing on multiple landscapes. We strongly feel that our approach of publishing the details of the method and our efforts to fully explore its capability are the most beneficial for the geomorphology community.

*I am, e.g., not sure whether the definition of χ was indeed introduced in the conference contribution by Royden et al. (2000) more than 10 years before it became popular; at least I did not find it in the cited abstract.*

This is stated in Perron and Royden (2013) on page 571, 2nd column, l.6-7: *The use of this coordinate transformation [referring to equation 6b exposing χ] to linearize river profiles was originally proposed by Royden et al. (2000) (···).*

*As a second example, the lower sections of page 5 read as if 2014 was more recent than 2017.*

We acknowledge the need for clarification here: the algorithm has been newly developed (2018) within TopoToolBox (citable as 2014) as the author (Wolfgang Schwanghart) details in the presentation of this feature (https://topotoolbox.wordpress.com/2018/06/29/finding-knickpoints-in-river-profiles/). Albeit unpublished (yet), we thought it was important to test our own method against others with the same goal of knickpoint extraction while taking completely different approaches. Wolfgang Schwanghart has also reviewed the paper and does not seem to object to our reference to the new method, but we agree that we should clarify the sequence of introduction of these tools. This has been done in section 1.1.2.

**Reviewer 2: Dr. Wolfgang Schwanghart**

We thank reviewer 2 (Wolfgang Schwanghart) for his thoughtful review of our manuscript. Below we list reviewer comments in italic font and our responses in regular font.

*Gailleton et al. present a method that automatically extracts knickpoints from longitudinal river profiles. The algorithms developed by the authors are well described and are implemented in LSD TopoTools, a terrain analysis software written and maintained by the authors. The algorithms are tested against hand-picked knickpoints and those derived with other software, and the code is publicly available. Overall, the manuscript is very well written and nicely illustrated. I have no concern about this paper being appropriate for the journal ESURF. To this end, I only have a few questions and some specific comments.*

Thanks. We are glad to hear the manuscript is clear.

*Would it make a difference, if you first smooth the elevation values using the TVD-approach and then calculate ksn? The smoothness-parameter would then be independent of theta.*

Thanks for this suggestion. We were not keen to do this in the first version of the manuscript as we didn't want to smooth elevation before searching for differences in χ gradient, since it would add an extra layer of complexity to the method. However, we have now attempted the smoothing of elevation using some different techniques to test if it makes a difference to the results.

Firstly, the TVD algorithm cannot be applied on the raw profile since it is designed to flatten signals and cannot be used on monotonically increasing data. Our approach has been to apply the TVD on a detrended elevation for each tributary (*i.e.,* applying the filter on $\Delta$ elevation rather than elevation itself. As shown on Figure 1, different values of $\lambda$ will generate different level of smoothing by flattening $\Delta z$ with different intensity. The denoised $\Delta z$ is then applied from the base level to the channel head to produce the denoised profiles. However, the denoising still depends on the $\lambda$ coefficient as the intensity of denoising might depend on the DEM quality and the user need (*e.g.,* focusing on large-scale gradient changes or small scale). Although the $\theta$ dependency cannot be avoided, we have added an elevation denoising option in the algorithm and also a description of this in the manuscript. In addition, we have added figures to the supplementary materials showing the performance of elevation denoising. However, we suggest to be cautious with adding denoising to the method as it involves data loss. We suggest

reading relevant literature (e.g., Schwanghart and Scherler, 2017) that discusses this specific issue before considering filtering initial dataset. The following figure illustrates the denoising results: a) shows the effect of lambda on the denoising intensity and b) the resulting $k_{sn}$ for denoising with a voluntary high regulation parameter $\lambda = 25$ to show that another denoising still is required. We mentioned and details these tests in the discussion of the main manuscript (section 5.1) and added these test results in the supplementary materials (Figures S18 to S20).

[Figure]

**Figure 1:** Effect of applying a TVD denoising filter on the elevation prior to the rest of the method as described in the manuscript. a) Long profile representation for values of $\lambda$. b)$k_{sn}$ calculated with Mudd et al. (2014) from a higly denoised profile ($\lambda = 25$). Even significantly denoised, it still requires a run of TVD to clean the signal. The noise magnitude is still dependent on $\theta$ in the same way described in the manuscript.

*Detecting knickpoints by identifying gradient-changes of ksn could also be achieved by calculating the profile curvature of the elevation data in chi-space. Similarly to $M_\chi$, this could be $C_\chi$ (or something similar). Of course, mathematically, this is the same. In addition, curvature is strongly affected by noise in the river long profile. However, using curvature instead of gradients of gradients is slightly more elegant and smoothing curvature might directly yield the peaks and troughs that you are looking for.*

Again, thanks for the suggestion. We tested several scenarios of curvature fitting to see if it improved our method. Tests suggested some potential for using curvature to detect and quantify knickpoints, however

there were several serious limitations. We first experimented $C_\chi = d(k_{sn})/d(\chi)$, where $k_{sn}$ is calculated with the segmentation algorithm Mudd et al. (2014) and filtered with the TVD. However as the discrete changes in $\chi$ between each node are quite variable, the resulting profile is significantly noisier than using $k_{sn}$ directly. The magnitude of each of the knickpoints detected with curvature becomes more sensitive to $\chi$ spacing and therefore $\theta$ compared to the method using $k_{sn}$. This is illustrated in Figure 2a where some of the $C_\chi$ differ from $\Delta k_{sn}$. $\theta$ in our case is relatively low, and therefore the discrete changes in $\chi$ happen to be in the same order of magnitude as the discrete changes in elevation. However, for higher values of $\theta$, $\Delta\chi$ can be several orders of magnitude lower than $\Delta z$ and therefore generate unnecessary high values. A similar issue is discussed in section 5.2) in the manuscript. Moreover, we find the $\Delta k_{sn}$ quantity more appealing as it can directly be translated into a drop/increase of channel slope (normalised to the concavity). We then explored the possibility of using a direct calculation of $\chi$- elevation profiles to detect knickpoints (*i.e.*, $d^2z/d\chi^2$). We applied a moving-average window on the $C_\chi$ and on $|C_\chi|$ to smooth and isolate peaks in curvature as suggested in the review. This method fails at identifying single outliers. Figure 2 shows the results of the three methods tested using curvature-based methods.

[Figure]

**Figure 2**: Methods based on chi curvature ($C_\chi$). a) The $C_\chi$ values (green) of a river reach derived from $C_\chi = d(k_{sn})/d(\chi)$ and compared with $\Delta k_{sn}$ values (in red) used in the manuscript to identify slope-break knickpoints. Their magnitude is similar but is very sensitive to $d\chi$, which is a function of $\theta$. The two circles show cases where $C_\chi$ and $\Delta k_{sn}$ show different and similar behaviour from the same original data. b) $C_\chi = d(k_{sn})/d(\chi)$ across the main river in the Smugglers Catchment. The red line represents a moving-average window of 20 nodes across the signal. The high and low peak values suggest some show potential to isolate knickpoints, but would require strong signal processing to be isolated: this method does not do better than our current method. c) Absolute value version of method b).

*Detecting change points in noisy data is a common topic in signal processing and statistics (see e.g. Truong et al., 2018). I wonder whether some of the techniques of knickpoint identification could actually be applied in a more formal statistical framework.*

Thanks for pointing out this reference; we have included it in the text. Alongside with this addition, we are adapting the vocabulary describing the method to fit with the statistical framework. Moreover, the reference offers (i) a review of the different statistical method to detect point changes and (ii) a python implementation of the main algorithm "*rupture*". The TVD suits our needs, but one might want a different method to adapt to a specific case study that would fall in the limitation of our method. We therefore adapted the code to generate raw files containing the output of the algorithm at different stages of our method. User can now, if needed, fit another method to ours using for example the "*rupture*" package. We (i) changed the section "Signal Denoising" to "Change point detection"

(section 2.3.1), (ii) linked our method to the broader statistical context of change point detection, (iii) adapted our vocabulary to fit the statistical terms and (iv) stated in section 5.4 that alternative change point detection can be used, pointing to relevant literature.

**Specific comments**

*6, 25: Filling might cause problems, because it can generate some large steps. Carving might be a better alternative.*

Yes, thanks for raising this. Our test examples were in locations where there were few roads and bridges. These features can generate steps after filling. The sites also had relatively little topographic noise and so we did not find valley filling to be a problem in our analysis. However, we recognise that many DEMs will have steps introduced by the filling algorithm. We therefore added a depression breaching algorithm in our software suite (Lindsay, 2016), as well as an option to directly feed the algorithm with a preprocessed raster from an alternative source (e.g., TopoToolbox, RichDEM). We have also added reference and small discussion to the carving algorithm in the text in section 2.1.

*8, 12: How much does "combining knickpoints" (2.3.2) actually affect the objective to identify the precise location of transitions between segments? It seems to me that knickpoint merging will let you pick knickzones, rather than knickpoints.*

We addressed this point by running a sensitivity analysis, which was available in the supplementary materials of the discussion materials (section 4.4). The segments are made of a large number of nodes and results show that, except for a large combining window (i.e., >100 nodes), the combining algorithm only cleans composite transitions between segments and does not combine large knickzones. We agree with the point that in the specific case of a close succession of knickpoints (e.g., a succession of waterfalls) and if the DEM precision is high enough to show them, then the algorithm might combine this succession of knickpoints as a single entity. We add this point to the main manuscript in section 5.1.

*Eq. 7: Denoising: The TVD algorithm (Eq. 7) is similar to the smoothing approach by Schwanghart and Scherler (2017), with the difference being the applied smoothness penalty. It would be interesting to know why you chose a gradient penalty instead of a curvature penalty. Wouldnt the gradient penalty require the horizontal distance in the denominator as the node-to-node distance may change depending on whether the node is a cardinal or diagonal neighbor?*

We developed a statistical approach in Mudd et al. (2014) to identify the best fit segments in chi-elevation space and the gradient is calculated on the basis of these segments: the spacing of the nodes is taken into account withing the segmentation routine. The TVD is then applied to the segmented data in order to minimise small variations in the already calculated $k_{sn}$ enabling extraction of knickpoint locations. We have added a reference to the study suggested above and described how it is different from our approach in the revised manuscript.

*12, 20: I was wondering about this error radius when reading through section 2.4. Consider to mention the radius also there. Did you use the same radius in the Brazilian test case?*

The radius has been chosen from Neely et al. (2017) based on their published parameters. We applied the same radius in the Brazilian case study (made with new field-derived dataset of knickpoint) for consistency. This has been clarified in the manuscript in section 4.2.

**Reviewer 3**

We thank reviewer 3 for their review of our manuscript. Below we list reviewer comments in italic font and our responses in regular font.

*This work presents an improved approach for the extraction and quantification of knickpoints from river long profiles. The work is well written and clear in its goals.*

Thanks.

*Having said that, in my opinion, there are two major issues to fix before getting it published: (1) the format of the publication that at my eyes is a technical note, not a full paper; (2) representativeness of study area considered in this study and its application in complex geomorphology landscapes.*

*(1) In the hands of the Esurf editor the decision, but at my eyes, this paper, mainly focused on an improved method, among others available in the literature, should be structured as a technical note.*

We disagree and have made our case in response to reviewer 1. We should also note that ESURF does not include a technical note category.

We repeat an excerpt from our response to reviewer 1:

Providing open-source and documented code is crucial for making our research easily reproducible, testable and improvable. However it also generates a significant risk of mis/over interpretations of its results, as the software would ultimately produce results in any context. We therefore believe that it is crucial to provide such analysis with extended discussion on its use and in comparison to other existing methods, sensitivity analysis on the different parameters, example of uses on different landscapes, and clear statements on how to constrain it. Other scientists who might come to use our methods should be aware of what the algorithms can provide and what the limitations might be. The reviewer suggests that we drop this analysis in the appendix of a case study. However, inadequate discussion of methodology along with failure to publish software is one of our major frustrations. We do not agree with the suggestion that we follow this approach. The reviewer then suggests we just publish an overview of the general software. This again we feel would be a major disservice to users of our methods since such a paper could never go into the details of each method; the development and testing of each method typically represents many months of effort, not to mention the many years of CPU time we devote to testing on multiple landscapes. We strongly feel that our approach of publishing the details of the method and our efforts to fully explore its capability are the most beneficial for the geomorphology community.

*(2) Im not sure, but are the study areas presented, enough to guarantee a robust analysis of the capability of the given method to work objectively in different landscape contexts, and complex morphological conditions? The impression (but maybe Im wrong) is that the landscape morphology of those areas is quite gentle...*

We do not know how the reviewer got the impression that the landscapes are quite gentle, given that

the Santa Cruz site has an overall channel slope approaching 0.1 (see Figure 7 in the discussion paper) and the Brazilian site has numerous waterfalls and knickzones that exceed 40 metres (see Figure 8 in the discussion paper). We have tested the method in many sites: there is not enough room to add all of them to the paper. We added an example of application in the supplementary materials (Figure S21) in a well-studied active tectonic environment (Inyo Range, CA, USA), using the global dataset SRTM to expand the range of example applications of our algorithm.

The Santa Cruz site features a range of bedrock types, tectonic forcings, and variable sea level. The Brazilian site features lithologic heterogeneity and its tectonic history is contested. We have noted these features in Sections 3.1 and 3.2 of the discussion paper. We are surprised the reviewer does not consider sites with lithologic, tectonic and base level heterogeneities in both space and time to be complex.

*For other points, Im generally in line with the feedback provided by reviewer 1.*

We have responded to reviewer 1 and refer to that response.

*References

[revised manuscript text omitted]

---

## Author Response (AR2)

**THE UNIVERSITY of EDINBURGH School of Geosciences**

Boris Gailleton School of Geosciences University of Edinburgh Drummond Street Edinburgh, EH8 9XP

Email: b.gailleton@sms.ed.ac.uk

Giulia Sofia Associate Editor, Earth Surface Dynamics

January 18, 2019

Dear Dr. Sofia,

Thank you for conducting the helpful and efficient review process that has greatly improved the quality of our manuscript. Below we list the AE and reviewer comments in italic font; our responses are in normal font.

Dear Authors, I received the reviewers' reports, and I believe the manuscript was improved after the first round of review. At this stage, one of the reviewers still highlights the need of a few more technical modifications to the paper, which should be addressed.

Details of the changes in response to technical comments are listed at the end of this response letter, as they all represent minor changes in the wording and corrections of typographical errors.

The other reviewer pointed out the importance of this work as a technical piece, improving our ability to investigate landscapes and specifically knickpoints. As you pointed out in your manuscript, the scientific community needs improved methods for knickpoints delineation, and I believe that your paper reports a significant technical advance in this context. The manuscript content presents a robust delineation method, rather than an enhanced theoryobservation on knickpoints (studies on knickpoints and their importance are well documented in the literature, as nicely underlined in the introduction of your manuscript). Due to its technical content, the paper actually well-fits the short communication type in ESurf, also considering that, if you wish, detailed and specific technical information such as your codes and samples for the research might be included as electronic supplements (although I see that they are currently already available through your GitHub repository). I would suggest to consider this option and frame the paper in this context, after addressing the technical modifications suggested by the reviewer.

We acknowledge the need for clarifications about our choice of publication type, as stated by referee 1 (Stefan Hergarten). We understand the reasoning behind the suggestion of changing the submission type regarding the technical nature of our work. As stated in the *manuscript types* description from the ESURF webpage, short communication format *should be short (a few pages only)*, and previously published examples are between 6 and 10 pages. Our use of the ESURF template suggests the current manuscript is around twice as long as a typical short communication. Thus, if we were to reformat the paper as a short communication, we would need to remove half the current manuscript and move it to the supplemental materials.

We are very reluctant to move the much of the paper to the supplemental materials: we have already placed information we believe is supplemental to the manuscript in that section and we believe what is in the current manuscript should be exposed in the main paper to the geomorphology community. These components include the theory behind the method, the choice of parameters, and the testing sites (which include, for the Brazilian site, new data). It is our opinion that moving any of these sections into the supplemental materials required to reformat as a *short communication* would jeopardise the overall quality of the manuscript. Essentially, we have struggled to identify sections that can actually fall into the already long supplemental materials without altering the manuscript quality.

Highly detailed and specific technical information such as computer programme code or user manuals can be included as electronic supplements.

Both the computer code and user manual are already available outside the manuscript (they are on the github pages). They are specifically written about the software operation rather than about the algorithm uses, reproducibility and limitations which are the scope of this manuscript. Our test sites are not samples used to illustrate the algorithm outputs, but studies specifically chosen to test the advantages and limitations of the method in various landscapes against (i) existing algorithms, (ii) field data (including an unpublished dataset). Critical discussion about the effect of each of the parameters is provided. Dropping this out of the main text would undermine the importance of carefully constraining the algorithm whereas we show that a wrong concavity, as one example amongst many, can lead to strong misinterpretations. Finally, we carefully compare our method with existing algorithms, which serves to frame the strengths and differences of each methods. We believe this justifies our choice of submission format and, in our opinion, a *short communication* about our algorithm would relegate the main text as a technical description of the method and a brief presentation of the outputs. As we stated in our first responses: the code will ultimately produce output, and we believe that shortening this contribution would be a disservice to the geomorphological community in the sense that none of the sections can be taken out without increasing the risk algorithm misuse.

Sincerely,

Boris Gailleton

**Technical comments from the associate editor**

I would also suggest some improvements, as I spotted some minor typos (i.e. caption of fig 10 knicpoint).

Thanks for spotting this, we fixed this typo and a similar one in the caption of table 1.

Also, I suggest using a different terminology rather than windowed standard deviation or windowed statistical approach, (i.e. standard deviation within the window and identification of the window for the statistical analysis of a node.

This is a good idea and makes the reading clearer, thanks. We slightly modified the section explaining this method (section 2.3.3) and adapted the caption of Fig. 5, which were the two part mentioning similar expression.

**Technical comments from reviewer 2 (Wolfgang Schwanghart)**

I thus have no further objections against publishing the manuscript as is. A few minor changes concern orthographic errors in the supplements to the paper. 4.6 Change heading to "Sensitivity to reference area" 4.7 Remove the colon at the end of the heading 5.2.1. Remove the colon at the end of the heading

Thanks for spotting these, we applied all these changes.

[revised manuscript text omitted]